# The neural correlates of novelty and variability in human decision-making under an active inference framework

**Shuo Zhang[1,2], Yan Tian[1], Quanying Liu[2]\*, Haiyan Wu[1]\***

[1]Centre for Cognitive and Brain Sciences and Department of Psychology, University of Macau, Macau, China; [2]Department of Biomedical Engineering, Southern University of Science and Technology, Shenzhen, China

## eLife Assessment

This **valuable** study addresses a central question in systems neuroscience (validation of active inference models of exploration) using a combination of behavior, neuroimaging, and modeling. The data provided offers **solid** evidence that humans do perceive, choose, and learn in a manner consistent with the essential ingredients of active inference, and that quantities that correlate with relevant parameters of this active inference scheme are encoded in different regions of the brain.

**\*For correspondence:**
liuqy@sustech.edu.cn (QL);
haiyanwu@um.edu.mo (HW)

**Competing interest:** The authors declare that no competing interests exist.

**Abstract** Active inference integrates perception, decision-making, and learning into a united theoretical framework, providing an efficient way to trade off exploration and exploitation by minimizing (expected) free energy. In this study, we asked how the brain represents values and uncertainties (novelty and variability), and resolves these uncertainties under the active inference framework in the exploration-exploitation trade-off. Twenty-five participants performed a contextual two-armed bandit task, with electroencephalogram (EEG) recordings. By comparing the model evidence for active inference and reinforcement learning models of choice behavior, we show that active inference better explains human decision-making under novelty and variability, which entails exploration or information seeking. The EEG sensor-level results show that the activity in the frontal, central, and parietal regions is associated with novelty, while the activity in the frontal and central brain regions is associated with variability. The EEG source-level results indicate that the expected free energy is encoded in the frontal pole and middle frontal gyrus and uncertainties are encoded in different brain regions but with overlap. Our study dissociates the expected free energy and uncertainties in active inference theory and their neural correlates, speaking to the construct validity of active inference in characterizing cognitive processes of human decisions. It provides behavioral and neural evidence of active inference in decision processes and insights into the neural mechanism of human decisions under uncertainties.

## Introduction

Active inference from the free energy principle provides a powerful explanatory tool for understanding the dynamic relationship between an agent and its environment (*Friston, 2010*). Free energy is a measure of an agent's uncertainty about the environment, which can be understood as the difference between the real environment state and the agent's estimated environment state (*Friston and Stephan, 2007*). In addition, expected free energy is the free energy about the future and can be used to guide the optimization process of decision-making. Under the active inference framework, perception, action, and learning are all driven by the minimization of free energy (*Figure 1*). By minimizing free

**Figure 1.** Active inference. (**a**) Qualitatively, agents receive observations from the environment and use these observations to optimize Bayesian beliefs under an internal cognitive (a.k.a., world or generative) model of the environment. Then agents actively sample the environment states by action, choosing actions that would make them in more favorable states. The environment changes its state according to agents' policies (action sequences) and transition functions. Then again, agents receive new observations from the environment. (**b**) From a quantitative perspective, agents optimize the Bayesian beliefs under an internal cognitive (a.k.a., world or generative) model of the environment by minimizing the variational free energy. Then agents select policies minimizing the expected free energy, namely, the surprise expected in the future under a particular policy.

energy, people can optimize decisions, which encompasses both the reduction of uncertainty about the environment (through *exploration*) and the maximization of rewards (through *exploitation*). Active inference (*Friston et al., 2016*) is a pragmatic implementation of the free energy principle to action, proposing that agents not only minimize free energy through perception but also through actions that enable them to reach preferable states. Briefly, in active inference, the agent has an internal cognitive model to approximate the hidden states of the environment (perception) and actively acts to enable oneself to reach preferable states (action) (see 'Contextual two-armed bandit task').

In recent years, the active inference framework has been applied to understanding cognitive processes and behavioral policies in human decisions. Many works provide support for the potential of the active inference framework to describe complex cognitive processes and give theoretical insights into behavioral dynamics (*Friston et al., 2017*; *Kirchhoff et al., 2018*; *Parr et al., 2022*; *Friston et al., 2009*). For instance, it is theoretically deduced in the active inference framework on the exploration and exploitation trade-off (*Schwartenbeck et al., 2019*; *Friston et al., 2016*), which trade-off is essential to the functioning of cognitive agents in many decision contexts (*O'Reilly and Tushman, 2011*; *Wilson et al., 2014*). Specifically, *exploration* is to take actions that offer extra information about the current environment, actions with higher uncertainty, while *exploitation* is to take actions to maximize immediate rewards given the current belief, actions with higher expected reward. The exploration-exploitation trade-off refers to an inherent tension between information (resolving uncertainty) and goal-seeking, particularly when the agent is confronted with incomplete information about the environment (*Gershman, 2019*). However, these theoretical studies have rarely been confirmed experimentally with lab empirical evidence from both behavioral and neural responses (*Friston, 2010*; *Friston and Stephan, 2007*). We aimed to validate the active framework in a decision-making task with electroencephalogram (EEG) neural recordings.

The decision-making process frequently involves grappling with varying forms of uncertainty, such as novelty – the kind of uncertainty that can be reduced through sampling, and variability – the inherent uncertainty (variance) presented by a stable environment. Studies have investigated these different forms of uncertainty in decision-making, focusing on their neural correlates (*Daw et al., 2006*; *Badre et al., 2012*; *Cavanagh et al., 2012*; *Payzan-LeNestour et al., 2013*). These studies utilized different forms of multi-armed bandit tasks, for example, the restless multi-armed bandit tasks (*Daw et al., 2006*; *Guha et al., 2010*), risky/safe bandit tasks (*Tomov et al., 2020*; *Fan et al., 2023*; *Payzan-LeNestour et al., 2013*), and contextual multi-armed bandit tasks (*Collins and Frank, 2018*; *Schulz et al., 2015*; *Collins and Frank, 2012*). However, these tasks only separate either variability from novelty in uncertainty or actions from states (perception). In our work, we develop a contextual multi-armed bandit task to enable participants to actively reduce novelty, avoid variability, and maximize rewards using various policies (see 'Behavioral results' and Figure 4a). Our task makes it

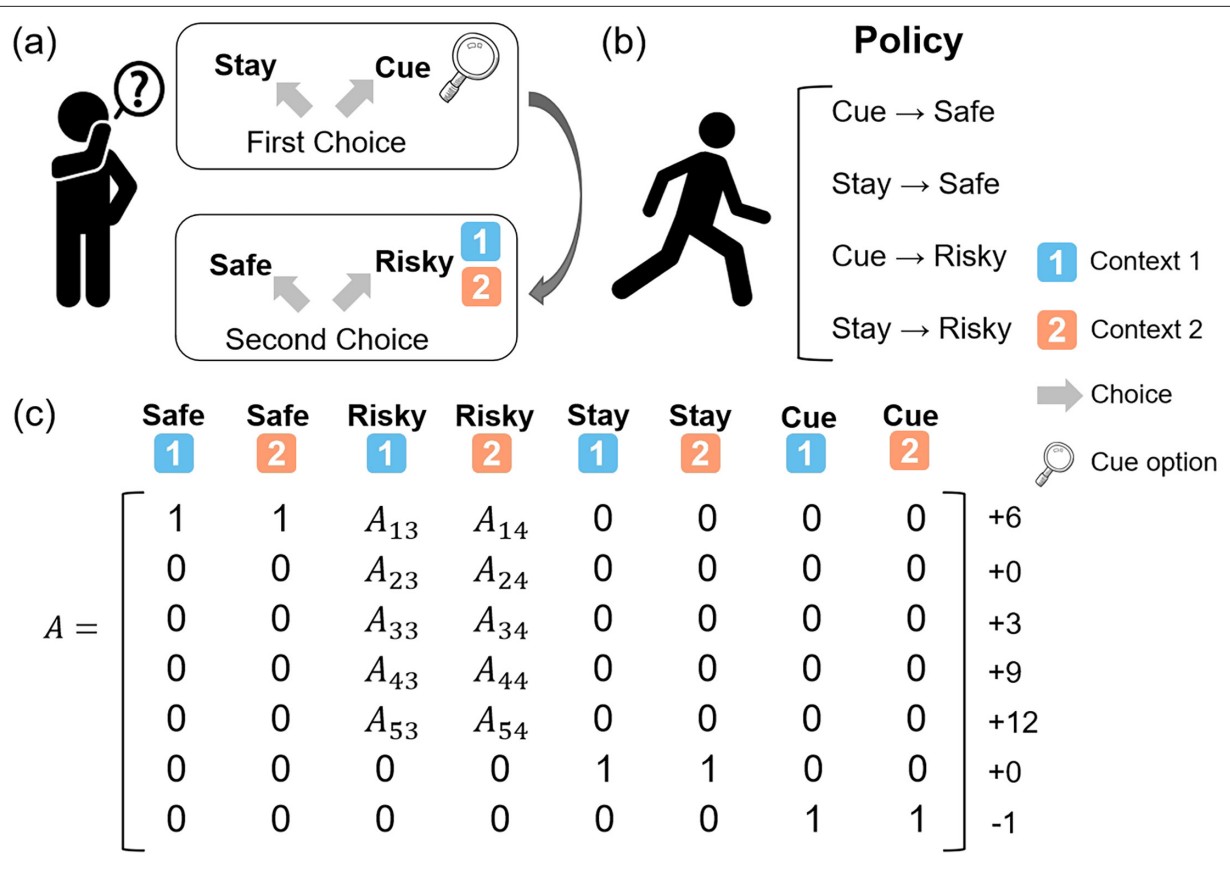

**Figure 2.** The contextual two-armed bandit task. (**a**) In this task, agents need to make two choices in each trial. The first choice is "Stay" and "Cue". The "Stay" option gives you nothing while the "Cue" option gives you a –1 reward and the context information about the "Risky" option in the current trial. The second choice is "Safe" and "Risky". The "Safe" option always gives you a +6 reward and the "Risky" option gives you a reward probabilistically, ranging from 0 to +12 depending on the current context (context 1 or context 2). (**b**) The four policies in this task are "Cue" and "Safe", "Stay" and "Safe", "Cue" and "Risky", and "Stay" and "Risky". (**c**) The likelihood matrix maps from 8 hidden states (columns) to 7 observations (rows).

possible to study whether the brain represents these different types of uncertainty distinctly (*Levy et al., 2010*) and whether the brain represents both the value of reducing uncertainty and the degree of uncertainty. The active inference framework presents a theoretical approach to investigate these questions. Within this framework, uncertainties can be reduced to novelty and variability. Novelty is represented by the uncertainty about model parameters associated with choosing a particular action, while variability is signified by the variance of the environment's hidden states. The value of reducing novelty, the value of reducing variability, and extrinsic value together constitute expected free energy (see 'Contextual two-armed bandit task').

Our study aims to utilize the active inference framework to investigate how the brain represents the decision-making process and how the brain disassociates the representations of novelty and variability (the degree of uncertainty and the value of reducing uncertainty). To achieve these aims, we utilize the active inference framework to examine the exploration-exploitation trade-off, with behavioral and EEG data (see 'Materials and methods'). Our study provides results of (1) how participants trade off the exploration and exploitation in the contextual two-armed bandit task (behavioral evidence), (2) how brain signals differ under different levels of ambiguities and risks (sensor-level EEG evidence), (3) how our brain encodes the trade-off of exploration and exploitation, evaluates the values of reducing novelty and reducing variability during action selection, and (4) updates the information about the environment during belief update (source-level EEG evidence).

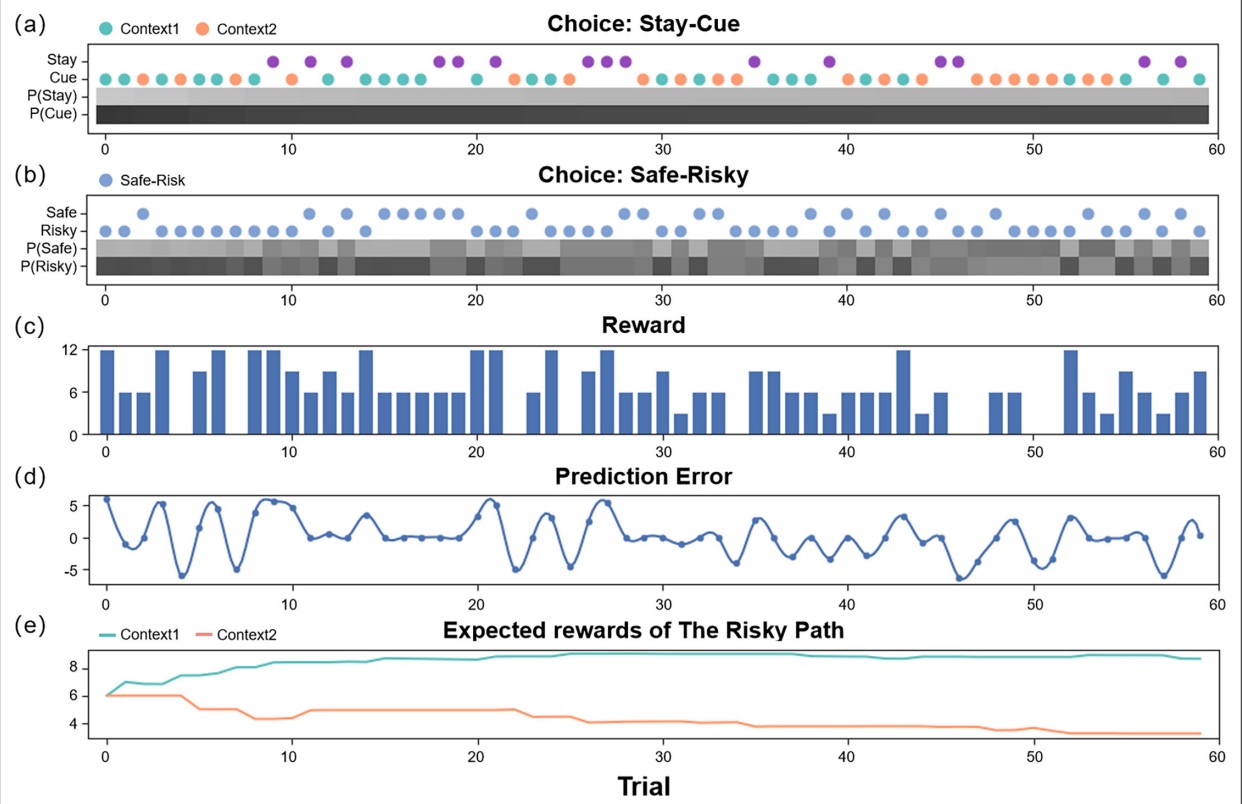

**Figure 3.** The simulation experiment results. This figure demonstrates how an agent selects actions and updates beliefs over 60 trials in the active inference framework. The first two panels (**a, b**) display the agent's policy and depict how the policy probabilities are updated (choosing between the stay or cue option in the first choice, and selecting between the safe or risky option in the second choice). The scatter plot indicates the agent's actions, with green representing the cue option when the context of the risky path is "Context 1" (high-reward context), orange representing the cue option when the context of the risky path is "Context 2" (low-reward context), purple representing the stay option when the agent is uncertain about the context of the risky path, and blue indicating the safe-risky choice. The shaded region represents the agent's confidence, with darker shaded regions indicating greater confidence. The third panel (**c**) displays the rewards obtained by the agent in each trial. The fourth panel (**d**) shows the prediction error of the agent in each trial, which decreases over time. Finally, the fifth panel (**e**) illustrates the expected rewards of the 'Risky Path' in the two contexts of the agent.

## Results

### Contextual two-armed bandit task

In this study, we developed a "contextual two-armed bandit task" (*Figure 2*), which was based on the conventional multi-armed bandit task (*Lu et al., 2010*; *Schwartenbeck et al., 2019*). Participants were instructed to explore two paths that offer rewards with the aim of maximizing cumulative rewards. One path provided constant rewards in each trial, labeled the "Safe", while the other, referred to as the "Risky", probabilistically offered varying amounts of rewards. The risky path had two different contexts, "Context 1" and "Context 2", each corresponding to different reward distributions. The risky path would give more rewards in "Context 1" and give fewer rewards in "Context 2". The context of the risky path changed randomly in each trial, and agents could only know the specific context of the current trial's risky path by accessing the "Cue" option, although this comes with a cost. The actual reward distribution of the risky path in "Context 1" was [+12 (55%), +9 (25%), +6 (10%), + 3(5%), + 0(5%)] and the actual reward distribution of the risky path in "Context 2" was [+12 (5%), +9 (5%), +6 (10%) + 3(25%) + 0 (55%)]. For a comprehensive overview of the specific settings, refer to *Figure 2*.

We ran some simulation experiments to demonstrate how active inference agents performed the "contextual two-armed bandit task" (*Figure 3*, *Appendix 1—figures 1 and 2*). Active inference agents with different parameter configurations could exhibit different decision-making policies, as demonstrated in the simulation experiment. By adjusting parameters such as *AL*, *AI*, *EX* (*Equation*

9), *prior* (*Equation 10*), and α (*Equation 11*), agents could operate under different policies. Agents with a low learning rate would initially incur a cost to access the cue, enabling them to thoroughly explore and understand the reward distributions of different contexts. Once sufficient environmental information was obtained, the agent would evaluate the actual values of various policies and select the optimal policy for exploitation. In the experimental setup, the optimal policy required selecting the risky path in a high-reward context and the safe path in a low-reward context after accessing the cue. However, in particularly difficult circumstances, an agent with a high learning rate might become trapped in a local optimum and consistently opt for the safe path, especially if the initial high-reward scenarios encountered yield minimal rewards.

*Figure 3* shows how an active inference agent with $AI = AL = EX = 1$ performs our task. We can see the active inference agent exhibits human-like policies and efficiency in completing tasks. In the early stages of the simulation, the agent tended to prefer the "Cue" option, as it provided more information, reducing novelty and reducing variability. Similarly, in the second choice, the agent favored the "Risky" option, even though initially the expected rewards for the "Safe" and "Risky" options were the same, but the "Risky" option offered greater informational value and reduced novelty. In the latter half of the experiment, the agent again preferred the "Cue" option due to its higher expected reward. For the second choice, the agent made decisions based on specific contexts, opting for the "Risky" option in "Context 1" for a higher expected reward, and the "Safe" option in "Context 2" where the informational value of the "Risky" option was outweighed by the difference in expected rewards between the "Safe" option and the "Risky" option in "Context 2".

In the behavioral experiment, in order to enrich the behavioral data of participants, a "you can ask" stage was added at the beginning of each trial. When the participants see "you can ask", they know that they can choose whether to ask for cue information in the next stage; when the participants see "you can't ask", they know that they can't choose whether to ask and it defaults that participants choose the "Stay" option. Additionally, to make the experiment more realistic, we added a background story of "finding apples" to the experiment. Specifically, participants were presented with the following instructions: *"You are on a quest for apples in a forest, beginning with 5 apples. You encounter two paths: (1) The left path offers a fixed yield of 6 apples per excursion. (2) The right path offers a probabilistic reward of 0/3/6/9/12 apples, and it has two distinct contexts, labeled 'Context 1' and 'Context 2,' each with a different reward distribution. Note that the context associated with the right path will randomly change in each trial. Before selecting a path, a ranger will provide information about the context of the right path ('Context 1' or 'Context 2') in exchange for an apple. The more apples you collect, the greater your monetary reward will be."*

The participants were provided with the task instructions (i.e., prior beliefs) above and asked to press a space bar to proceed. They were told that the total number of apples collected would determine the monetary reward they would receive. For each trial, the experimental procedure is illustrated in *Figure 4a* and comprises five stages:

1. "You can ask" stage: Participants are informed if they can choose to ask in the "First choice" stage. If they can't ask, it defaults that participants choose to not ask. This stage lasts for 2 seconds.
2. "First choice" stage: Participants decide whether to press the right or left button to ask the ranger for information, at the cost of an apple. In this stage, participants have 2 seconds to decide which option to choose, and they cannot press buttons within these 2 seconds. Then, they need to respond by pressing a button within another 2 seconds. This stage corresponds to the action selection in active inference.
3. "First result" stage: Participants either receive information about the context of the right path for the current trial or gain no additional information based on their choices. This stage lasts for 2 seconds and corresponds to the belief update in active inference.
4. "Second choice" stage: Participants decide whether to select the RIGHT or LEFT key to choose the safe path or risky path. In this stage, participants have two seconds to decide which option to choose, and they cannot press buttons within these 2 seconds. Then, they need to respond by pressing a button within another 2 seconds. This stage corresponds to the action selection in active inference.
5. "Second result" stage: Participants are informed about the number of apples rewarded in the current trial and their total apple count, which lasts for 2 seconds. This stage corresponds to the belief update in active inference.

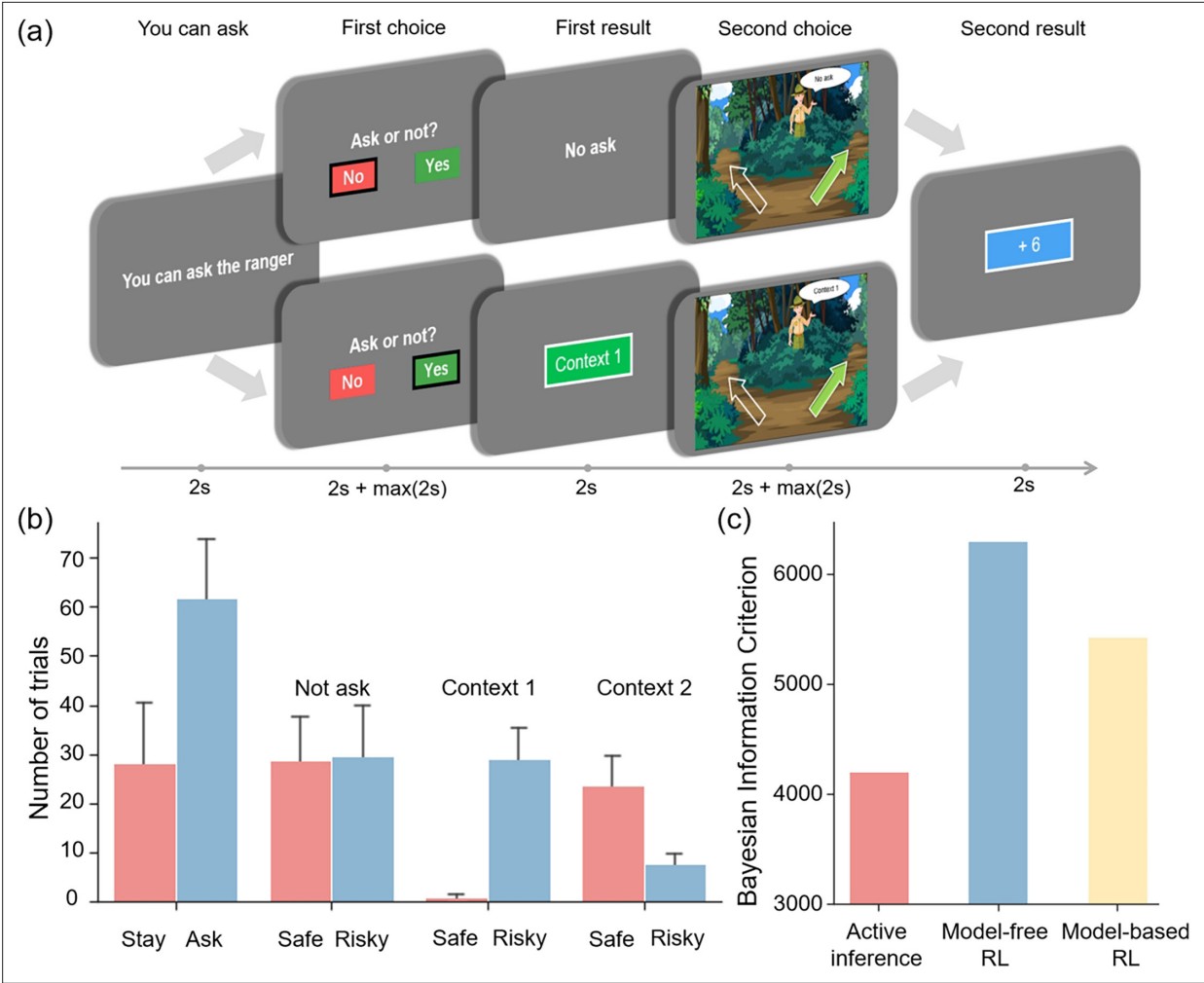

**Figure 4.** The experiment task and behavioral result. (**a**) The five stages of the experiment, which include the "You can ask" stage to prompt the participants to decide whether to request information from the Ranger, the "First choice" stage to decide whether to ask the ranger for information, the "First result" stage to display the result of the "First choice" stage, the "Second choice" stage to choose between left and right paths under different uncertainties and the "Second result" stage to show the result of the "Second choice" stage. The error bars show the 95% confidence interval. (**b**) The number of times each option was selected. The error bar indicates the variance among participants. (**c**) The Bayesian information criterion of active inference, model-free reinforcement learning, and model-based reinforcement learning.

Each stage was separated by a jitter ranging from 0.6 to 1.0 seconds. The entire experiment consisted of a single block with a total of 120 trials. The participants were required to use any two fingers of one hand to press the buttons (left arrow and right arrow on the keyboard).

## Behavioral results

To assess the evidence for active inference over reinforcement learning, we fitted active inference (*Equation 9*), model-free reinforcement learning, and model-based reinforcement learning models to the behavioral data of each participant. This involved optimizing the free parameters of active inference and reinforcement learning models. The resulting likelihood was used to calculate the Bayesian information criterion (BIC) (*Vrieze, 2012*) as the evidence for each model. The free parameters for the active inference model [$AL$, $AI$, $EX$, *prior* (*Equation 10*) and $\alpha$ *Equation 11*] scaled the contribution of the three terms that constituted the expected free energy in *Equation 9*. These coefficients could be regarded as precisions that characterized each participant's prior beliefs about contingencies and rewards. For example, increasing $\alpha$ meant participants would update their beliefs about reward contingencies more quickly, increasing $AL$ meant participants would like to reduce novelty more, and increasing $AI$ meant participants would like to learn the hidden state of the environment

and reduce variability more. The free parameters for the model-free reinforcement learning model were the learning rate α and the temperature parameter γ, and the free parameters for the model-based were the learning rate α, the temperature parameter γ and prior (the details for the model-free reinforcement learning model can be found in *Equations 12–22*, and the details for the model-based reinforcement learning model can be found in *Equations 23–34* in the supplementary method). The parameter fitting for these three models was conducted using the 'BayesianOptimization' package (*Frazier, 2018*) in Python, first randomly sampling 1000 times and then iterating for an additional 1000 times.

The model comparison results demonstrated that active inference provided a better performance to fit participants' behavioral data compared to the basic model-free reinforcement learning and model-based reinforcement learning (*Figure 4c*). Notably, the active inference could better capture the participants' exploratory inclinations (*Sutton and Barto, 2018*; *Friston et al., 2015*). This was evident in our experimental observations (*Figure 4b*) where participants significantly favored asking the ranger over opting to stay. Asking the ranger, which provided environmental information, emerged as a more beneficial policy within the context of this task.

Moreover, participants' preferences for information gain (i.e., epistemic value) were found to vary depending on the context. When participants lacked information about the context and the risky path had the same average rewards as the safe path but with greater variability, they showed an equal preference for both options (*Figure 4b*, "Not ask"). However, in "Context 1" (*Figure 4b*, high-reward context), where the risky path offered greater rewards than the safe path, participants strongly

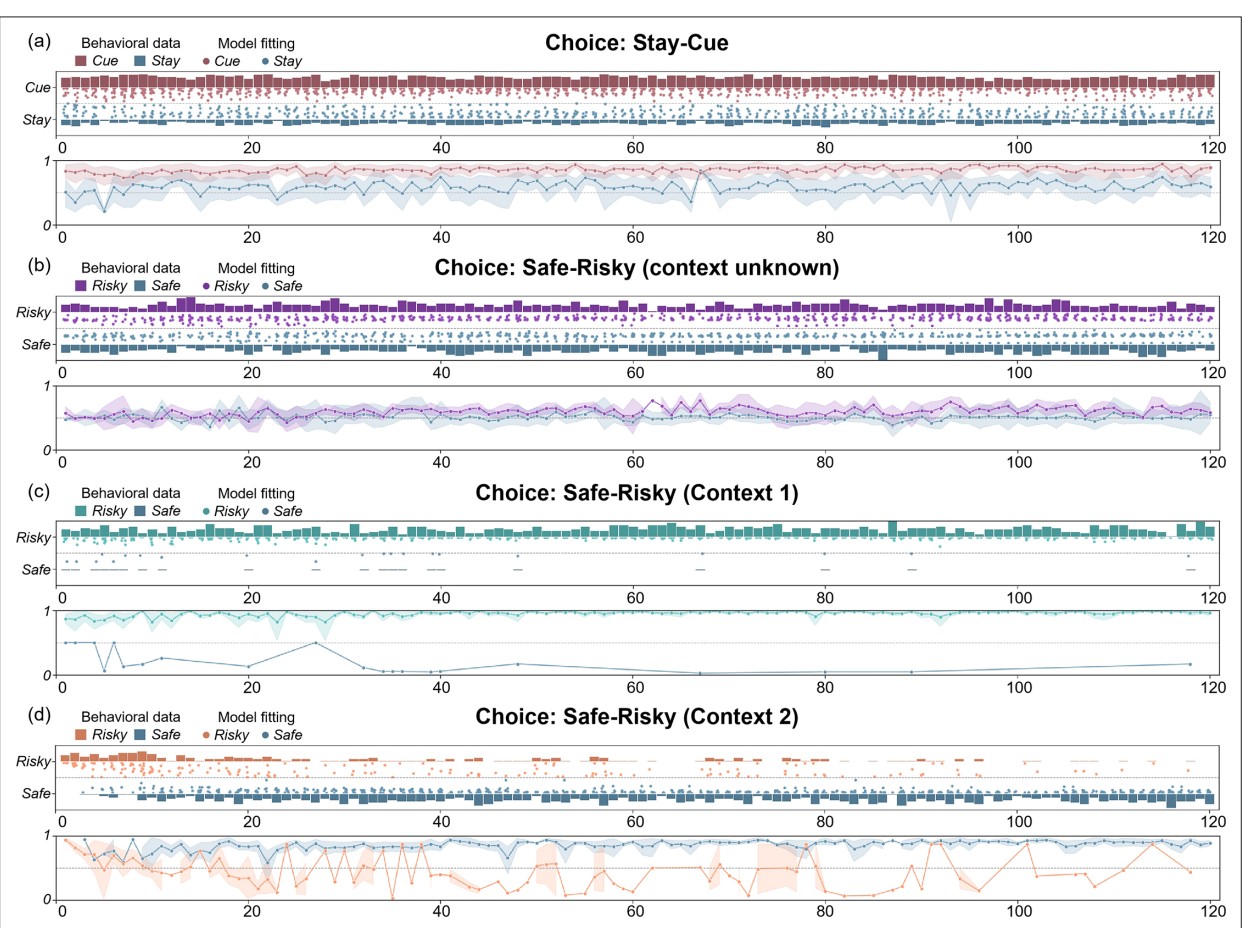

**Figure 5.** The comparison between the active inference model and the behavioral data in (**a**) the "First choice" stage, and the "Second choice" stage; (**b**) context unknown, (**c**) "Context 1", and (**d**) "Context 2". The bar graphs show participants' behavior data in each trial, and the height shows the proportion of participants who chose a certain option in each trial. The scatter plots show the model's fitting results for the two choices of the participants. The closer the point is to the bar graph on both sides, the higher the fitting accuracy. The line graphs show the trend of the model fitting accuracies with the trials.

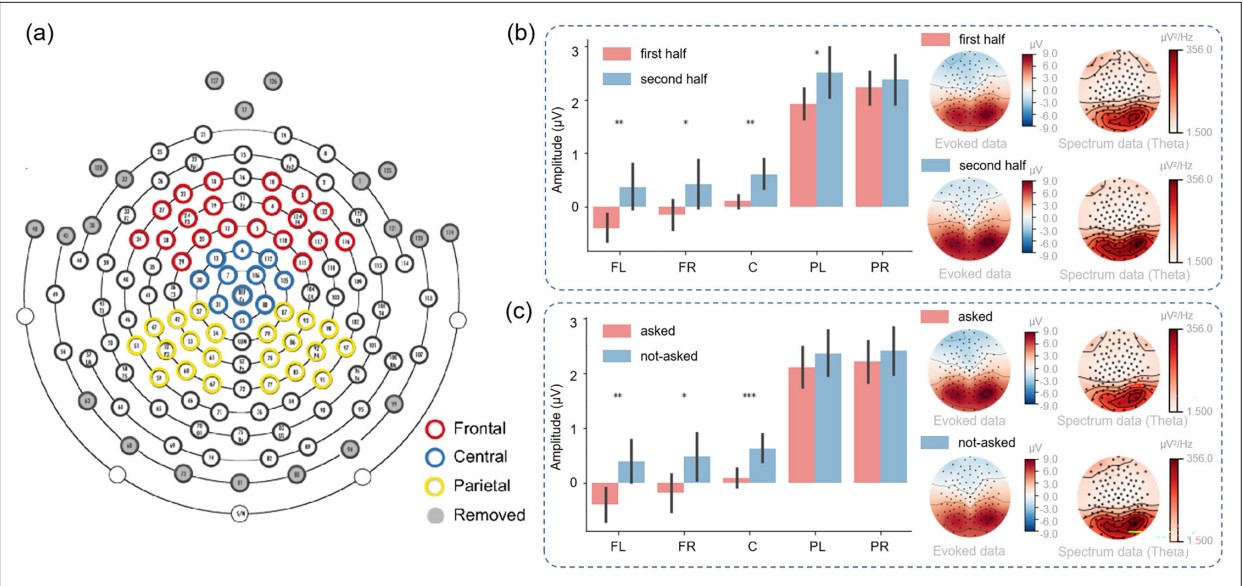

**Figure 6.** EEG results at the sensor level. (**a**) The electrode distribution. (**b**) The signal amplitude of different brain regions in the first and second half of the experiment in the "Second choice" stage. The error bar indicates the amplitude variance in each region. The right panel shows the visualization of the evoked data and spectrum data. (**c**) The signal amplitude of different brain areas in the "Second choice" stage where participants know the context or do not know the context of the right path. The error bar indicates the amplitude variance in each region. The error bars show the 95% confidence interval. The right panel shows the visualization of the evoked data and spectrum data. FL: frontal-left; FR: frontal-right; C: central; PL: parietal-left; PR: parietal-right.

favored the risky path, which not only provided higher rewards but also had more epistemic value. In contrast, in "Context 2" (**Figure 4b**, low-reward context), where the risky path had fewer rewards than the safe path, participants mostly chose the safe path but occasionally opted for the risky path, recognizing that despite its fewer rewards, it offered epistemic value.

**Figure 5** shows the comparison between the active inference model and the behavioral data, where we can see that the model can fit the participants' behavioral strategies well. In the "Stay-Cue" choice, participants always tend to choose to ask the ranger and rarely choose not to ask. When the context was unknown, participants chose the "Safe" option or the "Risky" option very randomly, and they did not show any aversion to variability. When given "Context 1", where the "Risky" option gave participants a high average reward, participants almost exclusively chose the "Risky" option, which provided more information in the early trials and was found to provide more rewards in the later rounds. When given "Context 2", where the "Risky" option gave participants a low average reward, participants initially chose the "Risky" option and then tended to choose the "Safe" option. We can see that participants still occasionally chose the "Risky" option in the later trials of the experiment, which the model does not capture. This may be due to the influence of forgetting. Participants chose the "Risky" option again to establish an estimate of the reward distribution.

## EEG results at sensor level

As depicted in **Figure 6a**, we divided electrodes into five clusters: left frontal, right frontal, central, left parietal, and right parietal. Within the "Second choice" stage, participants were required to make decisions under varying degrees of uncertainty (the uncertainty about the hidden states and the uncertainty about the model parameters). Thus, we investigated whether distinct brain regions exhibited different responses under such uncertainty.

In the first half of the experimental trials, participants would have greater uncertainty about model parameters compared to the latter half of the trials (**Schwartenbeck et al., 2019**). We thus analyzed data from the first half and latter half trials and identified statistically significant differences in the signal amplitude of the left frontal region ($p<0.01$), the right frontal region ($p<0.05$), the central region ($p<0.01$), and the left parietal region ($p<0.05$), suggesting a role for these areas in encoding the statistical structure of the environment (**Figure 6b**). We postulated that when participants had constructed

the statistical model of the environment during the second half of the trials, brains could effectively utilize the statistical model to make more confident decisions and exhibit greater neural responses.

To investigate whether distinct brain regions exhibited differential responses under the uncertainty about the hidden states, we divided all trials into two groups: the "asked" trials and the "not-asked" trials based on whether participants chose to ask in the "First choice" stage. In the *not-asked* (*Figure 6c*), participants had greater uncertainty about the hidden states of the environment compared to the *asked trials*. We identified statistically significant differences in the signal amplitude of the left frontal region (p<0.01), the right frontal region (p<0.05), and the central region (p<0.001), suggesting a role for these areas in encoding the hidden states of the environment. It might suggest that when participants knew the hidden states, they could effectively integrate the information with the environmental statistical structure to make more precise or confident decisions and exhibit greater neural response. The right panel of *Figure 6c* revealed a higher signal in the theta band during not-asked trials, suggesting a correlation between theta band signal and uncertainty about the hidden states (*Harper et al., 2017*).

## EEG results at source level

In the final analysis of the neural correlates of the decision-making process, as quantified by the epistemic and intrinsic values of expected free energy, we presented a series of linear regressions in source space. These analyses tested for correlations over trials between constituent terms in expected free energy (the value of reducing variability, the value of reducing novelty, extrinsic value, and expected free energy itself) and neural responses in source space. Additionally, we also investigated the neural correlate of (the degree of) variability, (the degree of) novelty, and prediction error. Because we were dealing with a 2-second time series, we were able to identify the periods of time during decision-making when the correlates were expressed. The linear regression was run by the " mne.stats.linear regression" function in the MNE package ($Activity \sim Regressor + Intercept$). *Activity* is the activity amplitude of the EEG signal in the source space and *regressor* is one of the regressors that we mentioned (e.g., expected free energy, the value of reducing novelty, etc.).

In these analyses, we focused on the induced power of neural activity at each time point, in the brain source space. To illustrate the functional specialization of these neural correlates, we presented whole-brain maps of correlation coefficients and picked out the brain region with the most significant correlation for reporting fluctuations in selected correlations over 2-second periods. These analyses were presented in a descriptive fashion to highlight the nature and variety of the neural correlates, which we unpacked in relation to the existing EEG literature in the discussion. The significant results after false discovery rate (FDR) (*Benjamini and Hochberg, 1995*; *Gershman et al., 2014*) correction are shown in shaded regions. Additional regression results can be found in supplementary materials.

### "First choice" stage: Action selection

During the "First choice" stage, participants were presented with the choice of either choosing to stay or ask the ranger to get information regarding the present context of the risky path, the latter choice coming at a cost. Here, we examined "expected free energy" ($G(\pi, \tau)$, *Equation 9*), "value of reducing variability" ($AI \cdot E_{\widetilde{Q}}[\ln Q(s_\tau|\pi) - \ln Q(s_\tau|o_\tau, \pi)]$), and "extrinsic value" ($EX \cdot E_{\widetilde{Q}}[\ln P(o_\tau)]$).

We found a robust correlation (p<0.05) between the "expected free energy" regressor and the frontal pole (*Figure 7a*). In addition, the superior temporal gyrus also displayed strong correlations with expected free energy. With respect to the "value of reducing variability" regressor, we identified a strong correlation (p<0.05) with the medial orbitofrontal cortex (*Figure 7b*). In addition, the post-central gyrus and precentral gyrus also displayed strong correlations with the value of reducing variability. For the "extrinsic value" regressor, we observed a strong correlation (p<0.05) with the middle temporal gyrus (see *Appendix 1—figure 4a*). In addition, the inferior temporal gyrus, and superior temporal gyrus also exhibited strong correlations with extrinsic value. Interestingly, we observed that during the "First choice" stage, expected free energy and extrinsic value regressors were both strongly correlated. However, expected free energy correlations appeared later than those of extrinsic value at the beginning, suggesting that the brain initially encoded reward values before integrating these values with information values for decision-making.

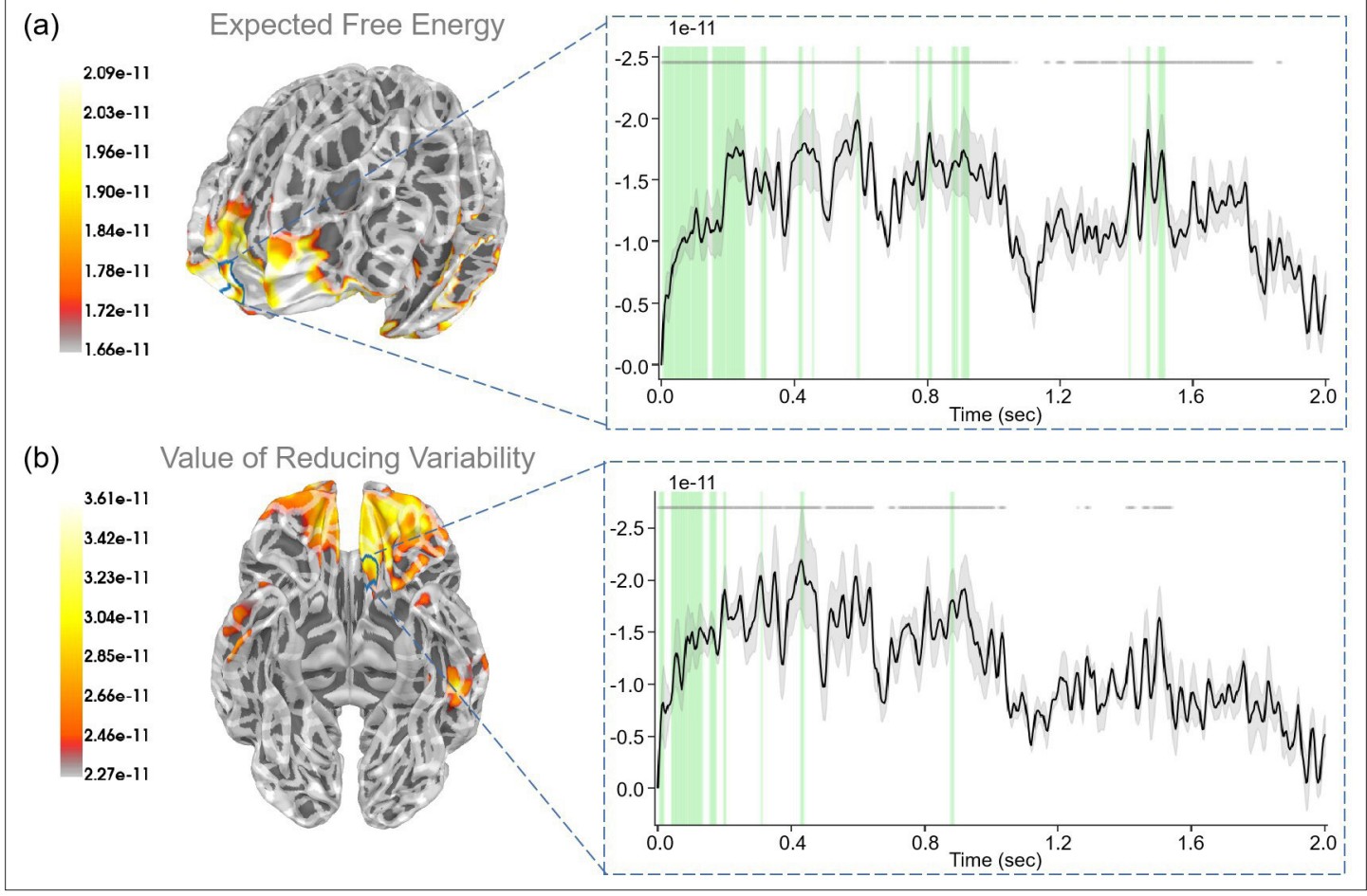

**Figure 7.** The source estimation results of expected free energy and active inference in the "First choice" stage. (**a**) The regression intensity ($\beta$) of expected free energy. The right panel indicates the regression intensity between the frontal pole (1, right half) and the expected free energy. The green-shaded regions indicate p<0.05 after false discovery rate (FDR) correction (the average *t*-value during these significant periods equals −3.228). (**b**) The regression intensity ($\beta$) of the value of reducing variability. The right panel indicates the regression intensity between the medial orbitofrontal cortex (5, left half) and the value of reducing variability. The green-shaded regions indicate p<0.05 after FDR correction (the average *t*-value during these significant periods equals −3.081). The black lines indicate the average intensities, and the gray-shaded regions indicate the ranges of variations (the 95% confidence interval). The gray lines indicate p<0.05 before FDR.

### "First result" stage: Belief update

During the "First result" stage, participants were presented with the outcome of their first choice, which informed them of the current context: either "Context 1" or "Context 2" for the risky path, or no additional information if they opted not to ask. This process correlated with the "reducing variability" regressor, as it corresponded to resolving uncertainties about hidden states. We assumed that the brain learning hidden states (reducing variability) corresponded to the value of reducing variability. Thus, the "reducing variability" regressor could be $AI \cdot (\ln Q(s_t|\pi) - \ln Q(s_t|o_t, \pi))$.

For "reducing variability", we observed a robust correlation (p<0.05) within the medial orbitofrontal cortex (**Figure 8a**). In addition, the rostral middle frontal gyrus, the lateral orbitofrontal cortex, and the superior temporal gyrus also displayed strong correlations with reducing variability.

### "Second choice" stage: Action selection

During the "Second choice" stage, participants chose between the risky path and the safe path based on the current information, with the aim of maximizing rewards. This required a balance between exploration and exploitation. Here, we examined "expected free energy" ($G(\pi, \tau)$, **Equation 9**), "value of reducing novelty" ($AL \cdot E_{\widetilde{Q}}[\ln Q(A) - \ln P(A|s_\tau, o_\tau, \pi)]$), "extrinsic value" ($EX \cdot E_{\widetilde{Q}}[\ln P(o_\tau)]$), and "novelty" ($E_{\widetilde{Q}}[\ln Q(A) - \ln P(A|s_\tau, o_\tau, \pi)]$).

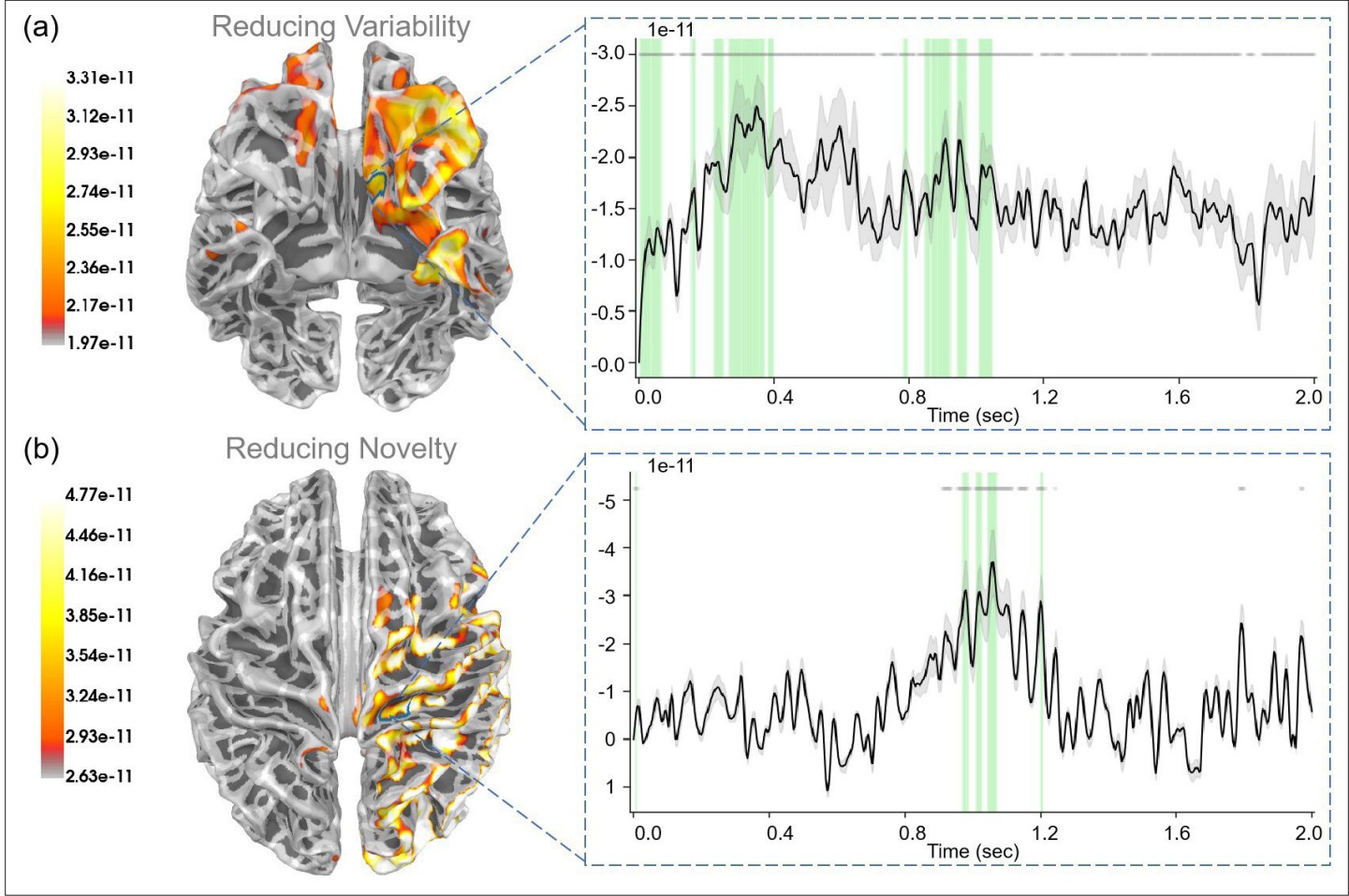

**Figure 8.** The source estimation results of reducing variability and reducing novelty in the two result stages. (**a**) The regression intensity (β) of reducing variability in the "First result" stage. The right panel indicates the regression intensity between the medial orbitofrontal cortex (5, left half) and reducing variability. The green-shaded regions indicate p<0.05 after false discovery rate (FDR) correction (the average $t$-value during these significant periods equals −3.001). (**b**) The regression intensity ($\beta$) of reducing novelty in the "Second result" stage. The right panel indicates the regression intensity between the precentral gyrus (15, right half) and reducing novelty. The green-shaded regions indicate p<0.05 after FDR correction (the average $t$-value during these significant periods equals −3.278). The black lines indicate the average intensities, and the gray-shaded regions indicate the ranges of variations (the 95% confidence interval). The gray lines indicate p<0.05 before FDR.

For "expected free energy" (*Figure 9a*), we identified strong correlations (p<0.001) in the rostral middle frontal gyrus. In addition, the caudal middle frontal gyrus, middle temporal gyrus, pars triangularis, and superior temporal gyrus also displayed strong correlations with expected free energy. Regarding the "value of reducing novelty", we found that the rostral middle frontal gyrus showed strong correlations (p<0.05). In addition, the superior frontal gyrus, insula, and lateral orbitofrontal cortex also displayed strong correlations with the value of reducing novelty. For "extrinsic value", strong correlations (p<0.001) were evident in the rostral middle frontal gyrus (see *Appendix 1—figure 4b*). In addition, the middle temporal gyrus, pars opercularis, and precentral gyrus also displayed strong correlations with extrinsic values. In the "Second choice" stage, participants made choices under different degrees of novelty. For "the degree of novelty", we found no significant correlations after FDR correction (see *Appendix 1—figure 6*). Generally, the correlations between regressors and brain signals were more pronounced in the "Second choice" stage compared to the "First choice" stage.

## "Second result" stage: Belief update

During the "Second result" stage, participants obtained specific rewards based on their second choice: selecting the safe path yields a fixed reward, whereas choosing the risky path results in

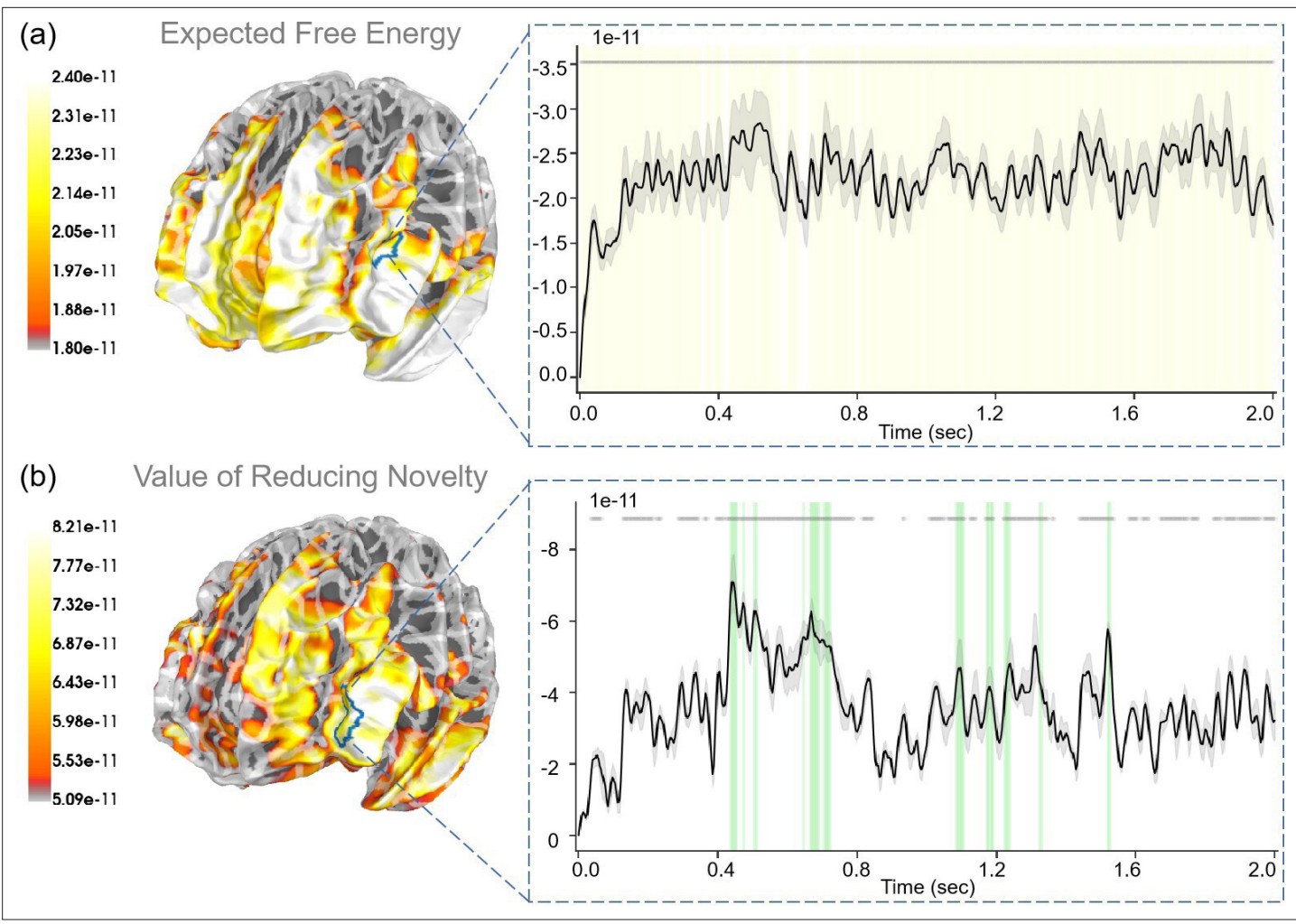

**Figure 9.** The source estimation results of expected free energy and the value of reducing novelty in the "Second choice" stage. (**a**) The regression intensity ($\beta$) of expected free energy. The right panel indicates the regression intensity between the rostral middle frontal gyrus (1, left half) and expected free energy, the black line indicates the average intensity of this region, and the gray-shaded region indicates the range of variation. The yellow-shaded regions indicate p<0.001 after false discovery rate (FDR) (the average $t$-value during these significant periods equals −4.819) and the gray lines indicate p<0.001 before FDR. (**b**) The regression intensity (β) of the value of reducing novelty. The right panel indicates the regression intensity between the rostral middle frontal gyrus (6, left half) and the value of reducing novelty, the black line indicates the average intensity of this region, and the gray-shaded region indicates the range of variation (the 95% confidence interval). The green-shaded regions indicate p<0.05 after FDR (the average $t$-value during these significant periods equals −3.067) and the gray lines indicate p<0.05 before FDR.

variable rewards, contingent upon the context. Here we examined "extrinsic value" ($r_t$), "prediction error" ($r_t - E_{\tilde{Q}}[\ln P(o_\tau)]$), and "reducing novelty" ($\ln Q(A) - \ln P(A|s_t, o_t, \pi)$). Here, we also assumed that learning model parameters (reducing novelty) corresponded to the value of reducing novelty.

For "extrinsic value", we observed strong correlations (p<0.05) in the lateral occipital gyrus, inferior parietal gyrus, and superior parietal gyrus (see **Appendix 1—figure 5a**). For "prediction error", we observed strong correlations (p<0.05) in the bank of the superior temporal sulcus, inferior temporal gyrus, and lateral occipital gyrus (see **Appendix 1—figure 5b**). For "reducing novelty", we observed strong correlations (p<0.05) in the precentral gyrus (see **Figure 8**).

## Discussion

In this study, we utilized active inference to explore the neural correlates involved in the human decision-making process under novelty and variability. By employing a contextual two-bandit task, we demonstrated that the active inference framework effectively describes real-world decision-making.

Our findings indicate that active inference not only provides explanations for decision-making under different kinds of uncertainty but also reveals the common and unique neural correlates associated with different types of uncertainties and decision-making policies. This was supported by evidence from both sensor-level and source-level EEG.

## The varieties of human exploration strategies in active inference

In the diverse realm of human behavior, it has been observed that human exploration strategies vary significantly depending on the current situation. Such strategies can be viewed as a blend of *directed exploration*, where actions with higher levels of uncertainty are favored, and *random exploration*, where actions are chosen at random (*Gershman, 2018*). In the framework of active inference, the randomness in exploration is derived from the precision parameter employed during policy selection. As the precision parameter increases, the randomness in agents' actions also increases. On the other hand, the directed exploration stems from the computation of expected free energy. Policies that lead to the exploration of more disambiguating options, hence yielding higher information gain, are assigned increased expected free energy by the model (*Friston et al., 2016*; *Friston et al., 2017*; *Gershman, 2019*).

Our model-fitting results indicate that people show high variance in their exploration strategies (*Figure 4b*). Exploration strategies, from a model-based perspective, incorporate a fusion of model-free learning and model-based learning. Intriguingly, these two learning ways exhibit both competition and cooperation within the human brain (*Gläscher et al., 2010*; *Daw et al., 2005*). The simplicity and effectiveness of model-free learning contrast with its inflexibility and data inefficiency. Conversely, model-based learning, although flexible and capable of forward planning, demands substantial cognitive resources. The active inference model tends to lean more toward model-based learning as this model incorporates a cognitive model of the environment to guide the agent's actions. Our simulation results showed these model-based behaviors in which the agent constructs an environment model and uses the model to maximize rewards (*Figure 3*). Active inference can integrate model-free learning through adding a habitual term (*Friston et al., 2016*). This allows the active inference agent to exploit the cognitive model (model-based) for planning in the initial task stages and utilize habits for increased accuracy and efficiency in later stages.

## The strength of the active inference framework in decision-making

Active inference is a comprehensive framework elucidating neurocognitive processes (*Figure 1*). It unifies perception, decision-making, and learning within a single framework centered around the minimization of free energy. One of the primary strengths of the active inference model lies in its robust statistical (*Crooks, 1998*) and neuroscientific underpinnings (*Lehmann et al., 2022*), allowing for a lucid understanding of an agent's interaction within its environment.

Active inference offers a superior exploration mechanism compared with basic model-free reinforcement learning (*Figure 4c*). Since traditional reinforcement learning models determine their policies solely on the state, this setting leads to difficulty in extracting temporal information (*Laskin et al., 2020*) and increases the likelihood of entrapment within local minima. In contrast, the policies in active inference are determined by both time and state. This dependence on time (*Wang et al., 2016*) enables policies to adapt efficiently, such as emphasizing exploration in the initial stages and exploitation later on. Moreover, this mechanism prompts more exploratory behavior in instances of state novelty. A further advantage of active inference lies in its adaptability to different task environments (*Friston et al., 2017*). It can configure different generative models to address distinct tasks and compute varied forms of free energy and expected free energy.

Despite these strengths, the active inference framework also has its limitations (*Raja et al., 2021*). One notable limitation pertains to its computational complexity (*Figure 2c*), resulting from its model-based architecture, restricting the traditional active inference model's application within continuous state-action spaces. Additionally, the model heavily relies on the selection of priors, meaning that poorly chosen priors could adversely affect decision-making, learning, and other processes (*Schwartenbeck et al., 2019*). However, sometimes it is just the opposite. As illustrated in the model comparison, priors can be a strength of Bayesian approaches. Under the complete class theorem (*Wald, 1947*; *Brown, 1981*), any pair of behavioral data and reward functions can be described in terms of ideal Bayesian decision-making with particular priors. In other words, there always exists a

description of behavioral data in terms of some priors. This means that one can, in principle, characterize any given behavioral data in terms of the priors that explain that behavior. In our example, these were effectively priors over the precision of various preferences or beliefs about contingencies that underwrite expected free energy.

## Representing uncertainties at the sensor level

The employment of EEG signals in decision-making processes under uncertainty has largely concentrated on event-related potential (ERP) and spectral features at the sensor level (*Wang et al., 2015*; *Lin et al., 2019*; *Bland and Schaefer, 2011*; *Botelho et al., 2023*). In our study, the sensor-level results reveal greater neural responses in multiple brain regions during the second half trials compared to the first half, and similarly, during not-asked trials as opposed to asked trials (*Figure 6*).

In our setting, after the first half of the trials, participants had learned some information about the environmental statistical structure, thus experiencing less novelty in the latter half of the trials. This increased understanding enabled them to better utilize the statistical structure for decision-making than they did in the first half of the trials. In contrast, during the not-asked trials, the lack of knowledge of the environment's hidden states led to higher-variability actions. This elevated variability was reflected in increased positive brain activities.

Novelty and variability, two pivotal factors in decision-making, are often misinterpreted and can vary in meaning depending on the context. Regarding the sensor level results, we find an overall greater neural response for the second half of the trials than the first half of the trials (*Figure 6b*). It may indicate a generally greater neural response for the lower novelty trials, which may contrast with previous studies showing greater neural response for higher novelty trials in previous studies (*Sun et al., 2017*; *Botelho et al., 2023*). For example, a late positive potential (LPP) was identified in their work, which differentiated levels of novelty, with the amplitude of the LPP serving as an index for perceptual novelty levels. However, the novelty in their task was defined as the perceptual difficulty of distinguishing, while our definition of novelty corresponds to the information gained from certain policies. Furthermore, *Zheng et al., 2020* used a wheel-of-fortune task to examine the ERP and oscillatory correlations of neural feedback processing under conditions of variability and novelty. Their findings suggest that risky gambling enhanced cognitive control signals, as evidenced by theta oscillation. In contrast, ambiguous gambling heightened affective and motivational salience during feedback processing, as indicated by positive activity and delta oscillation. Future work may focus on this oscillation level analysis and reveal more evidence on it.

## Representation of decision-making process in human brain

In our experiment, each stage corresponded to distinct phases of the decision-making process. Participants made decisions to optimize cumulative rewards based on current information about the environment during the two choice stages while acquiring information about the environment during the two result stages.

During the "First choice" stage, participants had to decide whether to pay an additional cost in exchange for information regarding the environment's hidden states. Here, the epistemic value stemmed from resolving the uncertainty about the hidden states and reducing variability. The frontal pole appears to play a critical role in this process by combining extrinsic value with epistemic value, expected free energy, to guide decision-making (*Figure 7*). Our results also showed the medial orbitofrontal cortex, postcentral gyrus, and precentral gyrus were correlated with the value of reducing variability. A previous study *Guo et al., 2013* demonstrated that the frontal pole was strongly activated in the risky condition and the ambiguous condition during decision-making. Another study also demonstrated that the frontal pole played an important role in the interaction between beliefs (variability and novelty) and payoffs (gains and losses) (*Smith et al., 2002*).

As for the "First result" stage, participants learned about the environment's hidden states and avoided risks in the environment. Our results indicated that the medial orbitofrontal cortex, rostral middle frontal gyrus, and lateral orbitofrontal cortex played a crucial role in both valuing the uncertainty about hidden states and learning information about these hidden states (*Figure 8a*). A previous study *Li et al., 2016* found that both the medial and lateral orbitofrontal cortex encoded variability and reward probability while the lateral orbitofrontal cortex played a dominant role in coding experienced value. Another study (*Rolls et al., 2022*) indicated that the medial orbitofrontal cortex was related to

risk-taking and risk-taking was driven by specific orbitofrontal cortex reward systems. Throughout the "First result" stage, participants are processing the state information relevant to the current trial. The orbitofrontal cortex is postulated to play a key role in processing this state information and employing it to construct an environmental model.

In the "Second choice" stage, participants chose between a safe path and a risky path based on their current information. When knowing the environment's hidden states, participants tended to resolve the uncertainty about model parameters by opting for the risky path. Conversely, without knowledge of the hidden states, Participants leaned toward variability reduction by choosing the safe path. Expected free energy is also correlated with brain signals but in different regions, such as the rostral middle frontal gyrus, caudal middle frontal gyrus, and middle temporal gyrus. Our results also highlighted the significance of the rostral middle frontal gyrus, superior frontal gyrus, insula, and lateral orbitofrontal cortex, in evaluating the value of reducing novelty. These results suggest that some brain regions may evaluate both the value of reducing novelty and the value of reducing variability (*Krain et al., 2006*).

For the "Second result" stage, participants got rewards according to their actions, constructing the value function and the state transition function. Our results highlighted the role of the precentral gyrus and superior parietal gyrus in learning the state transition function and reducing novelty (*Figure 8b*). Participants made their decisions in different contexts and there were multiple studies emphasizing the role of the superior parietal gyrus in uncertain decision-making (*Paulus et al., 2001*; *Huettel et al., 2005*; *Ragni et al., 2016*).

In the two "choice" stages, we observed stronger correlations for the expected free energy compared to the extrinsic value, suggesting that the expected free energy could better represent the actual value of the brain used to guide actions (*Williams et al., 2021*). Compared with the "First choice" stage, the correlations in the "Second choice" stage were more significant. This may indicate that the brain is activated more when making decisions for rewards than when making decisions for information. We found neural correlates for the value of reducing variability and the value of reducing novelty, but not the degree of variability and novelty (after FDR correction). Future work should design a task that highlights different degrees of variability and ambiguities. In the two result stages, the regression results of the "Second result" stage were not very reliable. This may be due to our discrete reward structure. Participants may not be good at remembering specific probabilities, but only the mean reward.

It should be acknowledged that only a subset of the regions identified without correction exhibit model parameter correlations that are robust enough to remain significant after correction for multiple comparisons. In future work, we should collect more precise neural data to make the results more robust, for example, collecting a head model for each subject instead of using an average model.

## Conclusion

In the current study, we introduce the active inference framework to investigate the neural mechanisms underlying an exploration and exploitation decision-making task. Compared to model-free reinforcement learning, active inference provides a superior exploration bonus and offers a better fit to the participants' behavioral data. Given that the behavioral task in our study only involved variables from a limited number of states and rewards, future research should strive to apply the active inference framework to more complex tasks. Specific brain regions may play key roles in balancing exploration and exploitation. The frontal pole and middle frontal gyrus were primarily involved in action selection (expected free energy). The precentral gyrus was mainly engaged in evaluating the value of reducing variability, and the rostral middle frontal cortex was also engaged in evaluating the value of reducing novelty. Furthermore, the medial orbitofrontal cortex participated in learning the hidden states of the environment (reducing variability) and the precentral gyrus participated in learning the model parameters of the environment (reducing novelty). In essence, our findings suggest that active inference is capable of investigating human behaviors in decision-making under uncertainty. Overall, this research presents evidence from both behavioral and neural perspectives that support the concept of active inference in decision-making processes. We also offer insights into the neural mechanisms involved in human decision-making under various forms of uncertainty.

## Materials and methods
### The free energy principle and active inference

The free energy principle (*Friston, 2010*) is a theoretical framework that proposes that both biological and non-biological systems tend to minimize their (variational) free energy to maintain a non-equilibrium steady state. In the context of the brain, the free energy principle suggests that the brain functions as an "inference machine" that aims to minimize the difference between its internal cognitive model about the environment and the true causes (hidden states) of perceived sensory inputs. This minimization is achieved through active inference.

Active inference can be regarded as a form of planning as inference, in which an agent samples the environment to maximize the evidence for its internal cognitive model of how sensory samples are generated. This is sometimes known as self-evidencing (*Friston et al., 2016*). Under the active inference framework, variational free energy can be viewed as the objective function that underwrites belief updating; namely, inference and learning. By minimizing the free energy expected following an action (i.e., expected free energy), we can optimize decisions and resolve uncertainty.

Mathematically, the minimization of free energy is formally related to variational Bayesian methods (*Galdo et al., 2020*). Variational inference is used to estimate both hidden states of the environment and the parameters of the cognitive model. This process can be viewed as an optimization problem that seeks to find the best model parameters and action policy to maximize the sensory evidence. By minimizing variational free energy and expected free energy, optimal model parameters can be estimated and better decisions can be made (*Friston, 2013*). Active inference bridges the sensory input, cognitive processes, and action output, enabling us to quantitatively describe the neural processes of learning about the environment. The brain receives sensory input $o$ from the environment, and the cognitive model encoded by the brain $q(s)$ makes an inference on the cause of sensory input $p(s|o)$ (a.k.a., the hidden state of the environment). In the free energy principle, minimizing free energy refers to minimizing the difference (e.g., KL divergence) between the cognitive model encoded by the brain and the causes of the sensory input. Thus, free energy is an information-theoretic quantity that bounds the evidence for the data model. Free energy can be minimized by the following two means (*Buckley et al., 2017*):

- Minimize free energy through *perception*. Based on existing observations, by maximizing model evidence, the brain improves its internal cognitive model, reducing the gap between the true cause of the sensory input and the estimated distribution of the internal cognitive model.
- Minimize free energy through *action*. The agent actively samples the environment, making the sensory input more in line with the cognitive model by sampling preferred states (i.e., prior preferences over observations). Minimizing free energy through action is one of the generalizations afforded by the free energy principle over Bayesian formulations that only address perception.

Active inference formulates the necessary cognitive processing as a process of belief updating, where choices depend on agents' expected free energy. Expected free energy serves as a universal

**Table 1.** Ingredients for computational modeling of active inference.

| Notations | Definition | Description |
|---|---|---|
| $O$ | A finite set of observations (outcomes) | Sensory input that brains receive. |
| $S$ | A finite set of hidden states | The true hidden states of the environment that generate sensory inputs to brains. |
| $U$ | A finite set of actions | Agent performs actions that change the environment. |
| $T$ | A finite set of time-sensitive policies | A policy is an action sequence over time. |
| $R$ | A generative process $R(\widetilde{o}, \widetilde{s}, \widetilde{u})$ | The generative process generates observations and next states (state transitions of the environment) based on current states and actions. |
| $P$ | A generative model $P(\widetilde{o}, \widetilde{s}, \pi, \eta)$ | The generative model describes what the agent believes about the environment (how observations are generated). |
| $Q$ | An approximate posterior | The Bayesian beliefs under the generative model that is optimized to minimize variational free energy. By definition, these beliefs correspond to approximate posteriors. |

objective function, guiding both perception and action. In brief, expected free energy can be seen as the expected surprise following some policies. The expected surprise can be reduced by resolving uncertainty, and one can select policies with lower expected free energy that can encourage information-seeking and resolve uncertainty. Additionally, one can minimize expected surprise by avoiding surprising or aversive outcomes (**Oudeyer and Kaplan, 2007**; **Schmidhuber, 2010**). This leads to goal-seeking behavior, where goals can be viewed as prior preferences or rewarding outcomes.

Technically, expected free energy can be expressed as risk plus ambiguity or in terms of expected information gain and expected value, where the value corresponds to (log) prior preferences. We will refer to both formulations in what follows. Resolving ambiguity (and maximizing information gain) has epistemic value, while avoiding risk (and maximizing expected value) has pragmatic or instrumental value. These two types of values can be referred to in terms of intrinsic and extrinsic value, respectively (**Barto et al., 2013**; **Schwartenbeck et al., 2019**).

## The generative model

Active inference builds on partially observable Markov decision processes: (O, S, U, T, R, P, Q) (see **Table 1**).

In this model, the generative model $P$ is parameterized as follows and the model parameters are $\eta = a, c, d, \beta$ (**Friston et al., 2016**):

$$P(\widetilde{o}, \widetilde{s}, \pi, \eta) = P(\pi)P(\eta) \prod_{t=1}^{T} P(o_t|s_t)P(s_t|s_{t-1}, \pi),$$
$$P(o_t|s_t) = Cat(A),$$
$$P(s_{t+1}|s_t, \pi) = Cat(B(\pi(t))),$$
$$P(s_0) = d, \tag{1}$$
$$P(\pi|\gamma) = \sigma(-\gamma \cdot G(\pi)),$$
$$P(A) = Dir(a),$$
$$P(\gamma) = \Gamma(1, \beta),$$

where $o$ is observations or sensory inputs ($\widetilde{o}$ is the history of observations), $s$ is the hidden states of the environment ($\widetilde{s}$ is the history of hidden states), $\pi$ is agent's policies, $A$ is the likelihood matrix mapping from hidden states to observations, $B$ is the transition function for hidden states under the policy in time $t$, $d$ is the prior expectation of each state at the beginning of each trial, $\gamma$ is the inverse temperature of beliefs about policies, $\beta$ is the prior expectation of policies' temperature parameters, $a$ is the concentration parameters of the likelihood matrix, $\sigma$ is the softmax function, $Cat()$ is the categorical distribution, $Dir()$ is the Dirichlet distribution, and $\Gamma()$ is the Gamma distribution.

The posterior probability of the corresponding hidden states and parameters ($x = \widetilde{s}, \pi, A, B, \beta$) is as **Equation 2**:

$$Q(x) = Q(s_1|\pi)...Q(s_T|\pi)Q(\pi)Q(A)Q(B)Q(\gamma) = \underset{Q(x)}{\arg\min} F \approx P(x|\widetilde{o}) \tag{2}$$

The generative model is a conceptual representation of how agents understand their environment. This model fundamentally posits that agents' observations are contingent upon states, and the transitions of these states inherently depend on both the state itself and the chosen policy. It is crucial to note that within this model the policy is considered a stochastic variable requiring inference, thus considering planning as a form of inference. This inference process involves deducing the optimal policy from the agents' observations. All the conditional abilities rest on likelihood and state transition models that are parameterized using a Dirichlet distribution (**FitzGerald et al., 2015**). The Dirichlet distribution's sufficient statistic is its concentration parameter, which is equivalently interpreted as the cumulative frequency of previous occurrences. In essence, this means that the agents incorporate the frequency of past combinations of states and observations into the generative model. Therefore, the generative model plays a pivotal role in inferring the probabilities and uncertainties related to the hidden states and observations.

## Variational free energy and expected free energy

Perception, decision-making, and learning in active inference are all achieved by minimizing the variational and expected free energy with respect to the model parameters and hidden states. The variational free energy can be expressed in various forms with respect to the reduced posterior as *Equation 3*:

$$
\begin{aligned}
F &= E_{Q(x)}[\ln Q(x) - \ln P(x, \widetilde{o})] \\
&= E_{Q(x)}[\ln Q(x) - \ln P(x|\widetilde{o}) - \ln P(\widetilde{o})] \\
&= E_{Q(x)}[\ln Q(x) - \ln P(\widetilde{o}|x) - \ln P(x)] \\
&= D_{KL}[Q(x)\|P(x)] - E_{Q(x)}[\ln P(\widetilde{o})]
\end{aligned}
\tag{3}
$$

Here, $x = \widetilde{s}, \pi, A, B, \beta$, including the hidden states and parameters. These forms of free energy are consistent with the variational inference in statistics. Minimizing free energy is equal to maximizing model evidence, that is, minimizing surprise. In addition, free energy can also be written in other forms as *Equation 4*:

$$
F = \underbrace{D_{KL}[Q(x)\|P(x)]}_{complexity} - \underbrace{E_{Q(x)}[\ln P(\widetilde{o}|x)]}_{accuracy}
\tag{4}
$$

The initial term, denoted as $D_{KL}[Q(x)\|P(x)]$, is conventionally referred to as "complexity". This term, reflecting the divergence between $Q(x)$ and $P(x)$, quantifies the volume of information intended to be encoded within $Q(x)$ that is not inherent in $P(x)$. The subsequent term, $E_Q[\ln P(\widetilde{o}|s)]$, designated as "accuracy", represents the likelihood of an observation expected under approximate posterior (Bayesian) beliefs about hidden states.

The minimization of variational free energy facilitates a progressive alignment between the approximate posterior distribution of hidden states, as encoded by the brain's cognitive function, and the actual posterior distribution of the environment. However, it is noteworthy that our policy beliefs are future-oriented. We want policies that possess the potential to effectively guide us toward achieving the future state that we desire. It follows that these policies should aim to minimize the free energy in the future, or in other words, expected free energy. Thus, expected free energy depends on future time points $\tau$ and policies $\pi$, and $x$ can be replaced by the possible hidden state $s_\tau$ and the likelihood matrix $A$. The relationship between policy selection and expected free energy is inversely proportional: a lower expected free energy under a given policy heightens the probability of that policy's selection. Hence, expected free energy emerges as a crucial ccice.

$$
\begin{aligned}
P(\pi) &= \sigma(-\gamma \cdot G(\pi)) \\
G(\pi) &= \sum_t G(\pi, \tau)
\end{aligned}
\tag{5}
$$

Next, we can derive the expected free energy in the same way as the variational free energy:

$$
G(\pi, \tau) = E_{\widetilde{Q}}[\ln Q(s_\tau, A|\pi) - \ln P(o_\tau, s_\tau, A|\pi)]
\tag{6}
$$

$$
\begin{aligned}
G(\pi, \tau) &= E_{\widetilde{Q}}[\ln Q(s_\tau, A|\pi) - \ln P(o_\tau, s_\tau, A|\pi)] \\
&= E_{\widetilde{Q}}[\ln Q(A) + \ln Q(s_\tau|\pi) - \ln P(A|s_\tau, o_\tau, \pi) - \ln P(s_\tau|o_\tau, \pi) - \ln P(o_\tau)] \\
&\approx E_{\widetilde{Q}}[\ln Q(A) + \ln Q(s_\tau|\pi) - \ln P(A|s_\tau, o_\tau, \pi) - \ln Q(s_\tau|o_\tau, \pi) - \ln P(o_\tau)]
\end{aligned}
\tag{7}
$$

In *Equation 7*, it is important to note that we anticipate observations that have not yet occurred. Consequently, we designate $\widetilde{Q} = Q(o_\tau, s_\tau, A|\pi)$. If we establish a relationship between $\ln P(o_\tau)$ and the prior preference, it enables us to express expected free energy in terms of epistemic value and extrinsic value. The implications of such a relationship offer a new lens to understand the interplay between cognitive processes and their environmental consequences, thereby enriching our understanding of decision-making under the active inference framework.

$$
G(\pi, \tau) = \underbrace{E_{\widetilde{Q}}[\underbrace{\ln Q(A) - \ln P(A|s_\tau, o_\tau, \pi)}_{novelty} + \underbrace{\ln Q(s_\tau|\pi) - \ln Q(s_\tau|o_\tau, \pi)}_{salience}]}_{Epistemic\ value} - \underbrace{E_{\widetilde{Q}}[\ln P(o_\tau)]}_{Extrinsic\ value}
\tag{8}
$$

In this context, extrinsic value aligns with the concept of expected utility. On the other hand, epistemic value corresponds to the anticipated information gain or the value of reducing uncertainty, encapsulating the exploration of both model parameters (novelty) and the hidden states (salience), which are to be illuminated by future observations. We can add coefficients ($AL$, $AI$, and $EX$) before these three terms of *Equation 8* to better simulate the diverse exploration strategies of agents:

$$G(\pi, \tau) = \underbrace{AL \cdot E_{\widetilde{Q}}[\ln Q(A) - \ln P(A|s_\tau, o_\tau, \pi)]}_{\text{Value of reducing novelty}} + \underbrace{AI \cdot E_{\widetilde{Q}}[\ln Q(s_\tau|\pi) - \ln Q(s_\tau|o_\tau, \pi)]}_{\text{Value of reducing variability}} - \underbrace{EX \cdot E_{\widetilde{Q}}[\ln P(o_\tau)]}_{\text{Extrinsic value}} \quad (9)$$

To align with different types of uncertainties and avoid conflicts with active inference terminology, the first two terms in *Equation 9* are referred to as the value of reducing novelty and variability, respectively, while the corresponding terms in *Equation 8* are termed novelty and variability.

Belief updates play a dual role by facilitating both inference and learning processes. The inference is here understood as the optimization of expectations about the hidden states. Learning, on the other hand, involves the optimization of model parameters. This optimization necessitates the finding of sufficient statistics of the approximate posterior that minimize the variational free energy. Active inference employs the technique of gradient descent to identify the optimal update method (*Friston et al., 2016*). In the present work, our focus is primarily centered on the updated methodology related to the likelihood mapping $A$ and the concentration parameter $a$ (rows correspond to observations, and columns correspond to hidden states):

$$a_0 = \begin{bmatrix} 1 & 1 & prior & prior & 0 & 0 & 0 & 0 \\ 0 & 0 & prior & prior & 0 & 0 & 0 & 0 \\ 0 & 0 & prior & prior & 0 & 0 & 0 & 0 \\ 0 & 0 & prior & prior & 0 & 0 & 0 & 0 \\ 0 & 0 & prior & prior & 0 & 0 & 0 & 0 \\ 0 & 0 & 0 & 0 & 1 & 1 & 0 & 0 \\ 0 & 0 & 0 & 0 & 0 & 0 & 1 & 0 \\ 0 & 0 & 0 & 0 & 0 & 0 & 0 & 1 \end{bmatrix}, \quad (10)$$

$$A = Cat(a),$$
$$a_{t+1} = a_1 + \alpha(o_t \otimes s_t) \quad (11)$$

## Participants

Participants were recruited via an online recruitment advertisement. We recruited 25 participants (male: 14, female: 11, mean age: 20.82 ± 2.12 years old), concurrently collecting EEG and behavioral data. All participants signed an informed consent form before the experiments. This study was approved by the local ethics committee of the University of Macau (BSERE22-APP006-ICI).

## EEG processing

The processing of EEG signals was conducted using the EEGLAB toolbox (*Martínez-Cancino et al., 2021*) in the MATLAB and the MNE package (*Esch et al., 2019*). The preprocessing of EEG data involved multiple steps, including data selection, downsampling, high- and low-pass filtering, and independent component analysis (ICA) decomposition. Two-second data segments were selected at various stages during each trial in *Figure 4a*. Subsequently, the data was downsampled to a frequency of 250 Hz and subjected to high- and low filtering within the 1–30 Hz frequency range. In instances where channels exhibited abnormal data, these were resolved using interpolation and average values. Following this, ICA was applied to identify and discard components flagged as noise.

After obtaining the preprocessed data, our objective was to gain a more comprehensive understanding of the specific functions associated with each brain region, mapping EEG signals from the sensor level to the source level. To accomplish this, we employed the head model and source space

available in the "fsaverage" of the MNE package. We utilized eLORETA (*Pascual-Marqui, 2007*) for mapping the EEG data to the source space and used the "aparc sub" parcellation for annotation (*Khan et al., 2018*).

We segmented the data into five intervals that corresponded to the five stages of the experiment. The first stage, known as the "You can ask" stage, informed participants whether they could ask the ranger. In the second stage, referred to as the "First choice" stage, participants decided whether to seek cues. The third stage, called the "First result" stage, revealed the results of participants' first choices. The fourth stage, known as the "Second choice" stage, involved choosing between choosing the safe or risky path. Finally, the fifth stage, named the "Second result" stage, encompassed receiving rewards. The 2 seconds in the two choosing stages when participants were thinking about which option to choose and the 2 seconds in the two result stages when the results were presented were used for analysis. Each interval lasted 2 seconds, and this categorization allowed us to investigate brain activity responses to different phases of the decision-making process. Specifically, we examined the processes of action selection and belief update within the framework of active inference.

## Acknowledgements

This work was mainly supported by the Science and Technology Development Fund (FDCT) of Macau [0127/2020/A3, 0041/2022/A], the Natural Science Foundation of Guangdong Province (2021A1515012509), Shenzhen-Hong Kong-Macao Science and Technology Innovation Project (Category C) (SGDX2020110309280100), MYRG of University of Macau (MYRG2022-00188-ICI), NSFC-FDCT Joint Program 0095/2022/AFJ, the SRG of University of Macau (SRG202000027-ICI), the National Key R&D Program of China (2021YFF1200804), National Natural Science Foundation of China (62001205), Shenzhen Science and Technology Innovation Committee (2022410129, KCXFZ2020122117340001), and Guangdong Provincial Key Laboratory of Advanced Biomaterials (2022B1212010003).

## Additional information

### Funding

| Funder | Grant reference number | Author |
| --- | --- | --- |
| The Science and Technology Development Fund (FDCT) of Macau | FDCT 0127/2020/A3 | Haiyan Wu |
| The Natural Science Foundation of Guangdong Province | 2021A1515012509 | Haiyan Wu |
| Shenzhen-Hong Kong-Macao Science and Technology Innovation Project | SGDX2020110309280100 | Haiyan Wu |
| MYRG of University of Macau | MYRG2022-00188-ICI | Haiyan Wu |
| NSFC-FDCT Joint Program | 0095/2022/AFJ | Haiyan Wu |
| The SRG of University of Macau | SRG202000027-ICI | Haiyan Wu |
| The National Key R&D Program of China | 2021YFF1200804 | Quanying Liu |
| National Natural Science Foundation of China | 62001205 | Quanying Liu |
| Shenzhen Science and Technology Innovation Committee | 2022410129 | Quanying Liu |

| Funder | Grant reference number | Author |
|--------|------------------------|--------|
| Guangdong Province Key Laboratory of Advanced Biomaterials | 2022B1212010003 | Quanying Liu |
| Shenzhen Science and Technology Innovation Committee | KCXFZ2020122117340001 | Quanying Liu |
| The Science and Technology Development Fund (FDCT) of Macau | FDCT 0041/2022/A | Haiyan Wu |

The funders had no role in study design, data collection and interpretation, or the decision to submit the work for publication.

## Author contributions

Shuo Zhang, Data curation, Formal analysis, Methodology, Writing – original draft, Writing – review and editing; Yan Tian, Data curation, Writing – original draft; Quanying Liu, Supervision, Methodology, Writing – original draft, Writing – review and editing; Haiyan Wu, Conceptualization, Supervision, Writing – original draft, Project administration, Writing – review and editing, Funding acquisition, Resources, Methodology

## Author ORCIDs

Shuo Zhang (i) http://orcid.org/0009-0002-1303-793X
Haiyan Wu (i) https://orcid.org/0000-0001-8869-6636

## Ethics

Human subjects: All participants signed an informed consent form before the experiments. This study was approved by the local ethics committee of the University of Macau (BSERE22-APP006-ICI).

Reviewer #1 (Public review): https://doi.org/10.7554/eLife.92892.4.sa1
Reviewer #3 (Public review): https://doi.org/10.7554/eLife.92892.4.sa2
Author response https://doi.org/10.7554/eLife.92892.4.sa3

# Additional files

## Supplementary files

MDAR checklist

## Data availability

All experiment data and analysis codes are available at GitHub: https://github.com/andlab-um/FreeEnergyEEG (copy archived at *Zhang, 2025*).

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

## Appendix 1

### Supplementary method

#### Model-free reinforcement learning

We used a model-free reinforcement learning approach as a baseline model. It involves two value functions, $Q_{Stay/Cue}$ and $Q_{Safe/Risk}$, which respectively describe the expected values of different options for the participants:

$$Q_{Stay/Cue} = \begin{bmatrix} Q_0 & Q_0 \end{bmatrix} \tag{12}$$

$$Q_{Safe/Risk} = \begin{bmatrix} Q_0 & Q_0 \\ Q_0 & Q_0 \end{bmatrix} \tag{13}$$

Here, $Q_0$ equals 6, corresponding to the fixed reward obtained for choosing the safe option. $Q_{Stay/Cue}(1)$ represents the total expected reward for choosing the "Stay" option in this round for the participant, while $Q_{Stay/Cue}(2)$ represents the total expected reward for choosing the "Cue" option in this round. $Q_{Safe/Risk}(1,1)$ and $Q_{Safe/Risk}(1,2)$ respectively denote the expected rewards for choosing the "Safe" option in "Context 1" and "Context 2", whereas $Q_{Safe/Risk}(2,1)$ and $Q_{Safe/Risk}(2,2)$ represent the expected rewards for choosing the "Risk" option in "Context 1" and "Context 2".

$$P_{Stay/Cue}(i) = \frac{e^{\gamma Q_{Stay/Cue}(i)}}{e^{\gamma Q_{Stay/Cue}(1)} + e^{\gamma Q_{Stay/Cue}(2)}} \tag{14}$$

Here, $i = 1$ corresponds to choosing the "Stay" option, and $i = 2$ corresponds to choosing the "Cue" option.

If participants decide whether to choose the "Safe" option or the "Risk" option when knowing the context ("Context 1"):

$$P_{Safe/Risk}(i) = \frac{e^{\gamma Q_{Safe/Risk}(i,1)}}{e^{\gamma Q_{Safe/Risk}(1,1)} + e^{\gamma Q_{Safe/Risk}(2,1)}} \tag{15}$$

If participants decide whether to choose the "afe" option or the "Risk" option when knowing the context ("Context 2"):

$$P_{Safe/Risk}(i) = \frac{e^{\gamma Q_{Safe/Risk}(i,2)}}{e^{\gamma Q_{Safe/Risk}(1,2)} + e^{\gamma Q_{Safe/Risk}(2,2)}} \tag{16}$$

If participants decide whether to choose the "Safe" option or the "Risk" option when without knowing the context:

$$P_{Safe/Risk}(i) = \frac{e^{\gamma (Q_{Safe/Risk}(i,1) + Q_{Safe/Risk}(i,2))/2}}{\sum_{j=1,2} e^{\gamma (Q_{Safe/Risk}(j,1) + Q_{Safe/Risk}(j,2))/2}} \tag{17}$$

After participants make two choices and receive rewards $r$, $Q_{Stay/Cue}$ and $Q_{Safe/Risk}$ are updated using delta learning rule (**Sutton and Barto, 1998**).

If participants chose the "Stay" option:

$$Q_{Stay/Cue}(1) = Q_{Stay/Cue}(1) + \alpha * (r - Q_{Stay/Cue}(1)); \tag{18}$$

If participants chose the 'Cue' option:

$$Q_{Stay/Cue}(1) = Q_{Stay/Cue}(1) + \alpha * ((r - 1) - Q_{Stay/Cue}(1)); \tag{19}$$

($-1$) is the cost of choosing the "Cue" option and α is the learning rate. $Q_{Safe/Risk}(1,1)$ and $Q_{Safe/Risk}(1,2)$ always equal 6 because choosing the "Safe" option consistently results in a fixed reward of 6.

If participants chose the "Risk" option when knowing the context ("Context 1"):

$$Q_{Safe/Risk}(2,1) = Q_{Safe/Risk}(2,1) + \alpha * (r - Q_{Safe/Risk}(2,1)); \tag{20}$$

If participants chose the "Risk" option when knowing the context ("Context 2"):

$$Q_{Safe/Risk}(2,2) = Q_{Safe/Risk}(2,2) + \alpha * (r - Q_{Safe/Risk}(2,2)); \tag{21}$$

If participants chose the "Risk" option when without knowing the context:

$$Q_{Safe/Risk}(2,1) = Q_{Safe/Risk}(2,1) + 0.5 * \alpha * (r - Q_{Safe/Risk}(2,1));$$
$$Q_{Safe/Risk}(2,2) = Q_{Safe/Risk}(2,2) + 0.5 * \alpha * (r - Q_{Safe/Risk}(2,2)). \tag{22}$$

## Model-based reinforcement learning

Here, we used a model-based reinforcement learning model as another baseline model. It involves the same likelihood matrix mapping from hidden states to outcomes $A$ and the concentration parameters of likelihood $a$.

$$a = \begin{bmatrix} 1 & 1 & prior & prior & 0 & 0 & 0 & 0 \\ 0 & 0 & prior & prior & 0 & 0 & 0 & 0 \\ 0 & 0 & prior & prior & 0 & 0 & 0 & 0 \\ 0 & 0 & prior & prior & 0 & 0 & 0 & 0 \\ 0 & 0 & prior & prior & 0 & 0 & 0 & 0 \\ 0 & 0 & 0 & 0 & 1 & 1 & 0 & 0 \\ 0 & 0 & 0 & 0 & 0 & 0 & 1 & 0 \\ 0 & 0 & 0 & 0 & 0 & 0 & 0 & 1 \end{bmatrix} \tag{23}$$

$$A = Cat(a) \tag{24}$$

When participants decide whether to choose the "Stay" option or the "Cue" option, they consider the expected values of the "Stay" option, the "Cue" option, the "Safe" option, and the "Risk" option under both "Context 1" and "Context 2":

$$Q_{Risk}^{Context1} = A[:,3] \cdot Preference^T, \tag{25}$$

$$Q_{Risk}^{Context2} = A[:,4] \cdot Preference^T, \tag{26}$$

$$Q_{Safe} = Q_{Safe}^{Context1} = Q_{Safe}^{Context2} = 6. \tag{27}$$

Here, *Preference* equals [6, 12, 9, 3, 0, 0, −1, −1]. If participants choose the "Stay" option, they will not know the context and make the decision based on the expected values of the "Safe" option and the "Risk" option:

$$Q_{Stay} = max(6, Q_{Risk}^{Context1}/2 + Q_{Risk}^{Context2}/2) \tag{28}$$

If participants choose the "Cue" option, they can then decide whether to choose the "Safe" option or the "Risk" option based on the context:

$$Q_{Cue} = \frac{max(6 - 1, Q_{Risk}^{Context1} - 1) + max(6 - 1, Q_{Risk}^{Context2} - 1)}{2} \tag{29}$$

Here, (−1) is the cost of choosing the "Cue" option. The probabilities of choosing the "Stay" option and the "Cue" option are, respectively:

$$P(Stay) = \frac{e^{\gamma Q_{Stay}}}{e^{\gamma Q_{Stay}} + e^{\gamma Q_{Cue}}},$$
$$P(Cue) = \frac{e^{\gamma Q_{Cue}}}{e^{\gamma Q_{Stay}} + e^{\gamma Q_{Cue}}}. \tag{30}$$

If participants decide whether to choose the "Safe" option or the "Risk" option when knowing the context ("Context 1"):

$$P(Safe) = \frac{e^{\gamma Q_{Safe}}}{e^{\gamma Q_{Safe}} + e^{\gamma Q_{Risk}^{Context1}}},$$
$$P(Risk) = \frac{e^{\gamma Q_{Risk}^{Context1}}}{e^{\gamma Q_{Safe}} + e^{\gamma Q_{Risk}^{Context1}}}.$$

(31)

If participants decide whether to choose the "Safe" option or the "Risk" option when knowing the context ("Context 2"):

$$P(Safe) = \frac{e^{\gamma Q_{Safe}}}{e^{\gamma Q_{Safe}} + e^{\gamma Q_{Risk}^{Context2}}},$$
$$P(Risk) = \frac{e^{\gamma Q_{Risk}^{Context2}}}{e^{\gamma Q_{Safe}} + e^{\gamma Q_{Risk}^{Context2}}}.$$

(32)

If participants decide whether to choose the "Safe" option or the "Risk" option when without knowing the context:

$$P(Safe) = \frac{e^{\gamma Q_{Safe}}}{e^{\gamma Q_{Safe}} + e^{\gamma (Q_{Risk}^{Context2} + Q_{Risk}^{Context1})/2}},$$
$$P(Risk) = \frac{e^{\gamma (Q_{Risk}^{Context2} + Q_{Risk}^{Context1})/2}}{e^{\gamma Q_{Safe}} + e^{\gamma (Q_{Risk}^{Context2} + Q_{Risk}^{Context1})/2}}.$$

(33)

The process of value updating is the same as the process of value updating in active inference:

$$a = a + \alpha \sum_{\tau} o_{\tau} \otimes s_{\tau}.$$

(34)

Here, α is the learning rate.

## Supplementary results

### Simulation results

The simulation experiments primarily focus on the utility of AI, AL, and EX in the active inference models, comparing the differences in decision-making among agents with varying AI, AL, and EX parameter values, considering their varying tendencies to avoid risk, reduce ambiguity, and maximize extrinsic rewards. The simulation experiments in the supplementary materials considered two situations $EX = 0$ and $AI = AI = 0$.

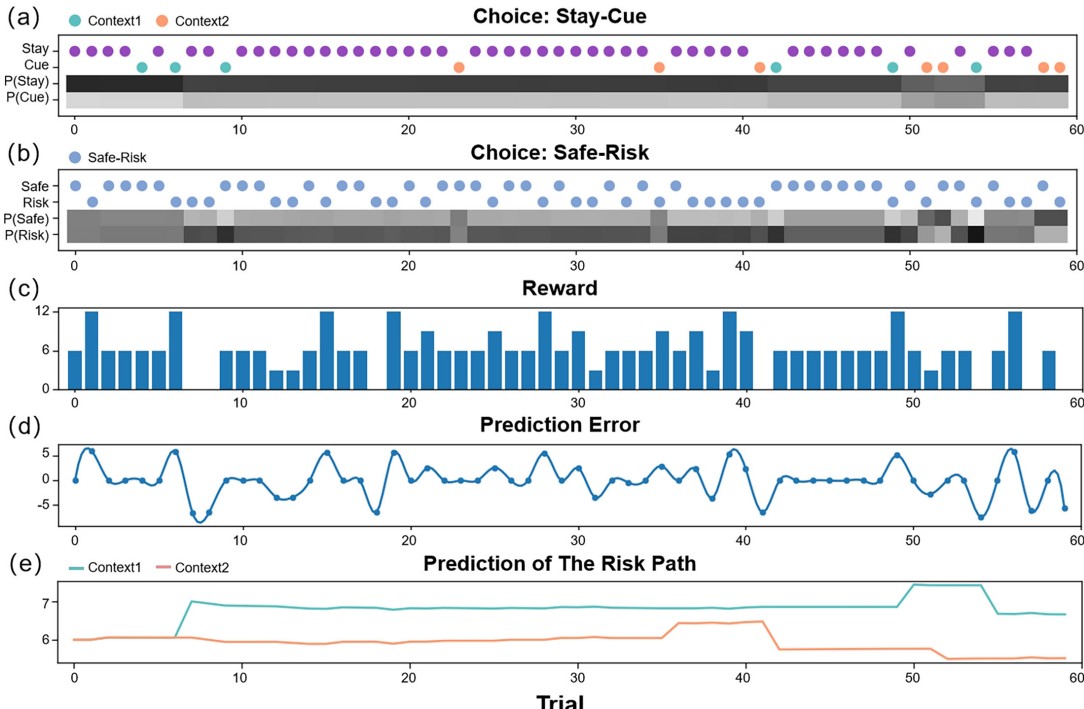

**Appendix 1—figure 1.** The simulation experiment results. This figure demonstrates how an agent selects actions and updates beliefs over 60 trials in the active inference framework. The first two panels (**a**, **b**) display the agent's policy and depict how the policy probabilities are updated (choosing between the stay or cue option in the first choice, and selecting between the safe or risky option in the second choice). The scatter plot indicates the agent's actions, with green representing the cue option when the context of the risky path is "Context 1" (high-reward context), orange representing the cue option when the context of the risky path is "Context 2" (low-reward context), purple representing the stay option when the agent is uncertain about the context of the risky path, and blue indicating the safe-risky choice. The shaded region represents the agent's confidence, with darker shaded regions indicating greater confidence. The third panel (**c**) displays the rewards obtained by the agent in each trial. The fourth panel (**d**) shows the prediction error of the agent in each trial. Finally, the fifth panel (**e**) illustrates the expected rewards of the "Risky Path" in the two contexts of the agent.

When AI and AL are set to 0 and EX is set to 10, it is expected that the agent will solely maximize extrinsic rewards without engaging in exploration. *Appendix 1—figure 2* depicts the decisions of such an agent. In the initial trials, the agent almost exclusively selects the "Stay" option, choosing the "Cue" option only occasionally, as the expected rewards for the "Risk" and "Safe" options are both 6 initially, and choosing the "Cue" option involves a cost. Without a bias toward either the "Safe" or "Risk" option, the agent's second choice is random. In the latter half, as the agent occasionally selects the "Cue" and "Risk" options, updating its expectations of the "Risk" option's reward, it increases the probability of choosing the "Cue" option. For the second choice, the agent eventually learns the optimal strategy, selecting the "Risk" option in "Context 1" and the "Safe" option in "Context 2". While the agent with AI and AL set to 0 can eventually learn the optimal strategy, its rate of strategy optimization is significantly slower compared to an agent with non-zero values for all three parameters.

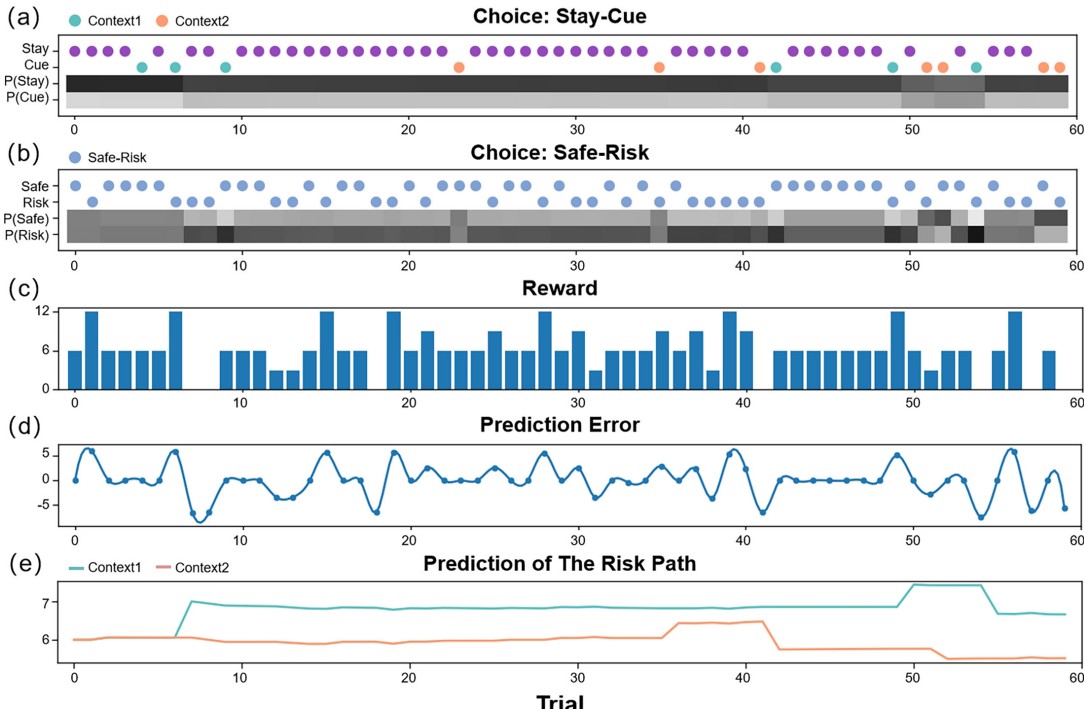

**Appendix 1—figure 2.** The simulation experiment results. This figure demonstrates how an agent selects actions and updates beliefs over 60 trials in the active inference framework. The first two panels (**a, b**) display the agent's policy and depict how the policy probabilities are updated (choosing between the stay or cue option in the first choice, and selecting between the safe or risky option in the second choice). The scatter plot indicates the agent's actions, with green representing the cue option when the context of the risky path is "Context 1" (high-reward context), orange representing the cue option when the context of the risky path is "Context 2" (low-reward context), purple representing the stay option when the agent is uncertain about the context of the risky path, and blue indicating the safe-risky choice. The shaded region represents the agent's confidence, with darker shaded regions indicating greater confidence. The third panel (**c**) displays the rewards obtained by the agent in each trial. The fourth panel (**d**) shows the prediction error of the agent in each trial. Finally, the fifth panel (**e**) illustrates the expected rewards of the "Risky Path" in the two contexts of the agent.

If EX is set to 0, and AI and AL are set to 10, the agent is expected to solely maximize information gain without pursuing higher extrinsic rewards. *Appendix 1—figure 1* illustrates the decisions of such an agent. Regardless of the experiment's stage, the agent overwhelmingly prefers the "Cue" option in the first choice, as it consistently offers the same value in avoiding risk. Similarly, in the second choice, the agent favors the "Risk" option due to its ability to reduce ambiguity. The agent explores the "Risk" option equally in both "Context 1" and "Context 2". However, it is noteworthy that in the latter half of the experiment, the agent's probability of selecting the "Safe" option increases, as continued exploration of the "Risk" option decreases its perceived ambiguity, rendering its informational value negligible.

In comparison, the active inference model exhibits an excellent exploration mechanism, enabling the agent to quickly learn optimal strategies in uncertain environments. Disabling the model's AL and AI parameters by setting them to 0 undermines this exploration mechanism, forcing the agent to optimize its strategy slowly through random exploration.

## Model recovery

To demonstrate how reliable our models are (the active inference model, model-free reinforcement learning model, and model-based reinforcement learning model), we run some simulation experiments for model recovery. We use these three models, with their own fitting parameters, to generate some simulated data. Then we will fit all three sets of data using these three models. The model recovery results are shown in *Appendix 1—figure 3*. This is the confusion matrix of models: the percentage of all subjects simulated based on a certain model that is fitted best by a certain model. The goodness of fit was compared using the Bayesian information criterion. We can see that

the result of model recovery is very good, and the simulated data generated by a model can be best explained by this model.

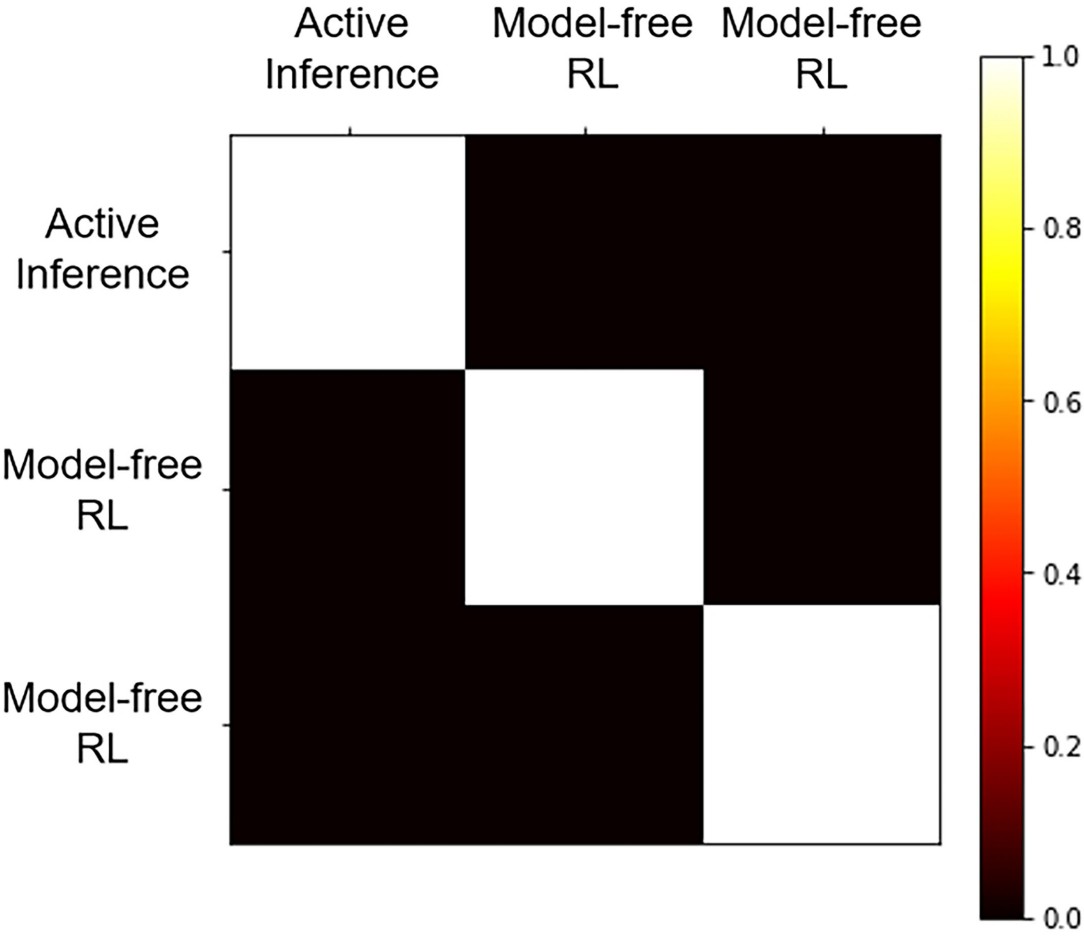

**Appendix 1—figure 3.** Model recovery results.

## Source estimate results

In our results analysis, different criteria lead to different outcomes. For instance, requiring a certain proportion of brain regions to exhibit significant correlations, specifying whether the significance level should be p<0.05 or p<0.01, or mandating a minimum duration for significant correlations in a certain proportion of brain regions all yield distinct results. Therefore, we compiled all brain regions that meet various criteria in the supplementary materials.

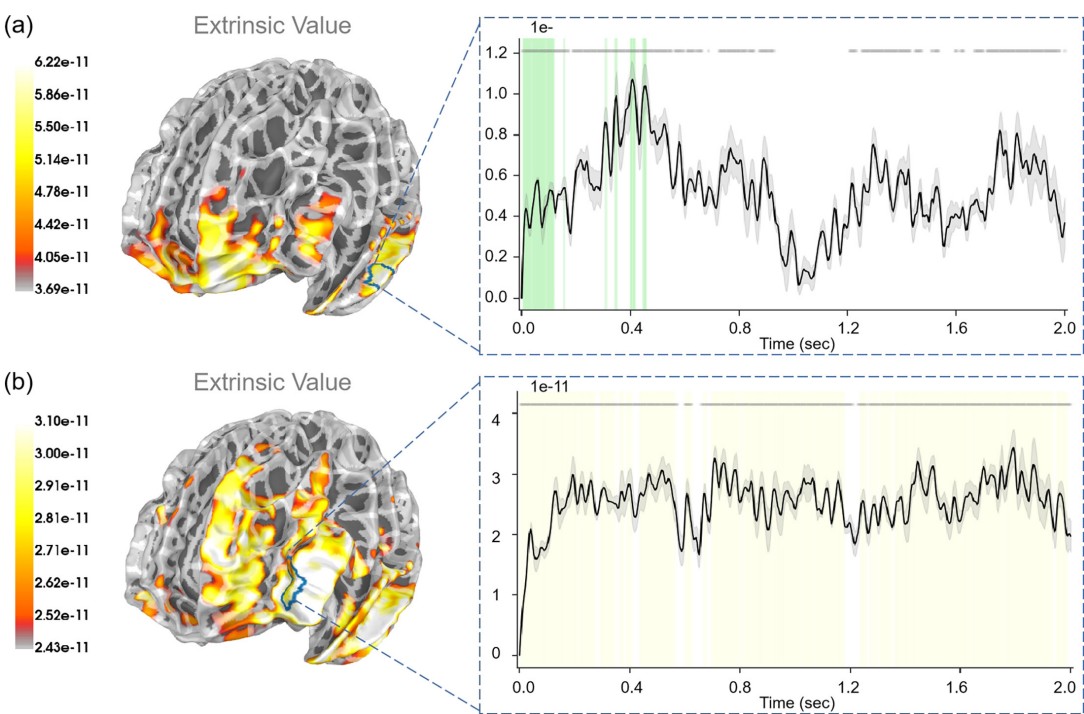

**Appendix 1—figure 4.** The source estimation results of extrinsic value in the two choosing stages. (**a**) The regression intensity ($\beta$) of extrinsic value in the "First choice" stage. The right panel indicates the regression intensity between the middle temporal gyrus (6, right half) and extrinsic value. The green-shaded regions indicate p<0.05 after false discovery rate (FDR) correction (the average *t*-value during these significant periods equals 3.673). (**b**) The regression intensity ($\beta$) of extrinsic value in the "Second choice" stage. The right panel indicates the regression intensity between the rostral middle frontal gyrus (6, left half) and extrinsic value. The yellow-shaded regions indicate p<0.001 after FDR correction (the average *t*-value during these significant periods equals 4.740). The black lines indicate the average intensities, and the gray-shaded regions indicate the ranges of variations. The gray lines indicate p<0.05 before FDR.

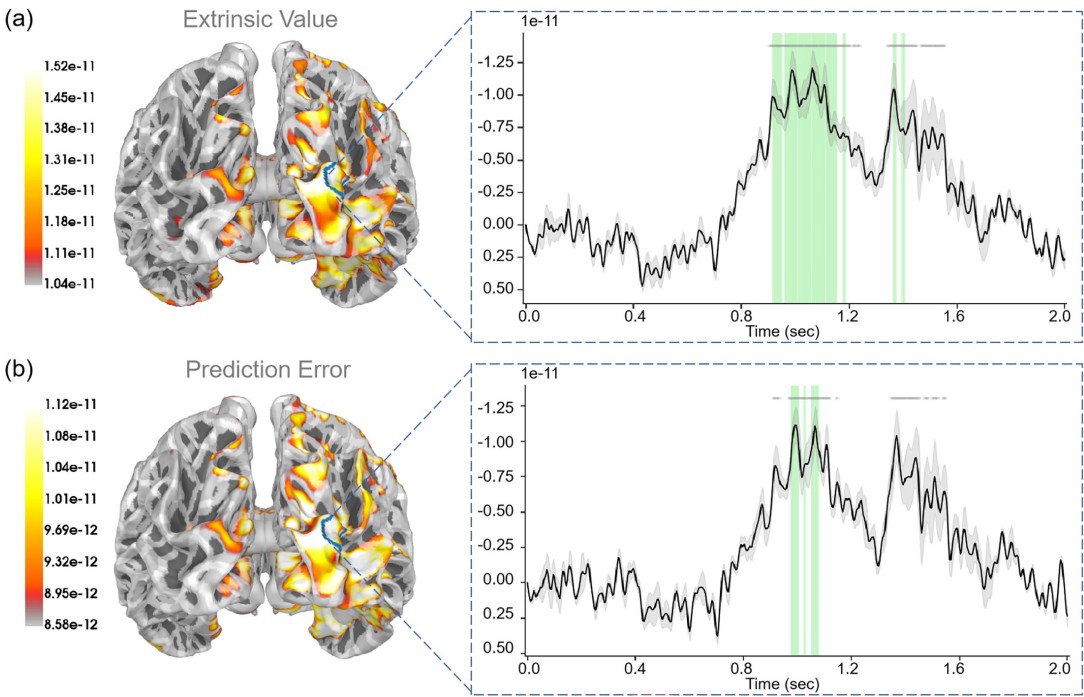

**Appendix 1—figure 5.** The source estimation results of extrinsic value and prediction error in the "Second result" stage. (**a**) The regression intensity ($\beta$) of extrinsic value. The right panel indicates the regression intensity between the lateral occipital cortex (3, right half) and extrinsic value. The green-shaded regions indicate p<0.05 after false discovery rate (FDR) correction (the average *t*-value during these significant periods equals 2.875). (**b**) The regression intensity ($\beta$) of prediction error. The right panel indicates the regression intensity between the lateral occipital cortex (3, right half) and prediction error. The green-shaded regions indicate p<0.05 after FDR correction (the average *t*-value during these significant periods equals –2.716). The black lines indicate the average intensities, and the gray-shaded regions indicate the ranges of variations. The gray lines indicate p<0.05 before FDR.

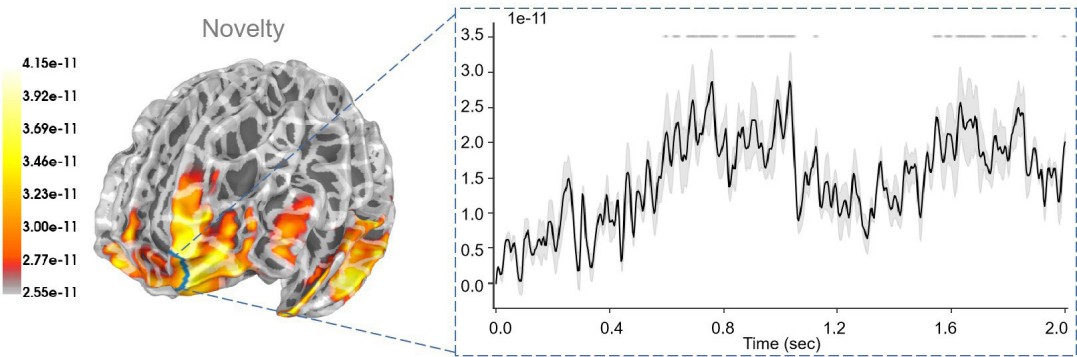

**Appendix 1—figure 6.** The source estimation results of ambiguity in the "Second choice" stage. The right panel indicates the regression intensity between the frontal pole (1, left half) and ambiguity. The black line indicates the average intensities, and the gray-shaded regions indicate the ranges of variations. The gray lines indicate p<0.05 before FDR.

**Appendix 1—table 1.** In the "Stay/Cue" choice, we require that the activity of more than 50% of brain regions remains significantly correlated with the expected free energy for over 0.32 seconds, with a significance p<0.05 (after false discovery rate correction).
The brain regions are delineated according to the "aparc sub" parcellation.

| Regressor | p-Value | Proportion | Duration | |
|---|---|---|---|---|
| Expected free energy | 0.01 | 0.5 | 0.32 seconds | |
| **Brain region** | **Duration** | **Proportion** | **Regression coefficient** | ***t*-Value** |
| Frontalpole 1-lh | 0.34 | 0.8965 | $1.342 \times 10^{-11}$ | −3.136 |
| Frontalpole 1-rh | 0.372 | 0.9086 | $1.357 \times 10^{-11}$ | −3.228 |
| Lateralorbitofrontal 2-lh | 0.364 | 0.8608 | $1.526 \times 10^{-11}$ | −3.235 |
| Lateralorbitofrontal 3-lh | 0.332 | 0.7691 | $1.362 \times 10^{-11}$ | −3.195 |
| Lateralorbitofrontal 5-lh | 0.336 | 0.7778 | $1.365 \times 10^{-11}$ | −3.126 |
| Lateralorbitofrontal 5-rh | 0.332 | 0.8193 | $1.333 \times 10^{-11}$ | −2.984 |
| Lateralorbitofrontal 6-lh | 0.372 | 0.9086 | $1.550 \times 10^{-11}$ | −3.335 |
| Lateralorbitofrontal 7-lh | 0.332 | 0.8614 | $1.584 \times 10^{-11}$ | −3.062 |
| Lateralorbitofrontal 7-rh | 0.34 | 0.8902 | $1.530 \times 10^{-11}$ | −3.059 |
| Medialorbitofrontal 1-lh | 0.356 | 0.9711 | $1.624 \times 10^{-11}$ | −3.068 |
| Medialorbitofrontal 1-rh | 0.348 | 0.9234 | $1.563 \times 10^{-11}$ | −3.049 |
| Medialorbitofrontal 2-lh | 0.348 | 0.9163 | $1.360 \times 10^{-11}$ | −3.038 |
| Medialorbitofrontal 3-rh | 0.336 | 0.9048 | $1.402 \times 10^{-11}$ | −3.016 |
| Medialorbitofrontal 4-lh | 0.364 | 0.8767 | $1.436 \times 10^{-11}$ | −3.165 |
| Medialorbitofrontal 5-lh | 0.38 | 0.8579 | $1.502 \times 10^{-11}$ | −3.420 |
| Rostralmiddlefrontal 11-lh | 0.364 | 0.8773 | $1.604 \times 10^{-11}$ | −3.200 |
| Rostralmiddlefrontal 11-rh | 0.344 | 0.8469 | $1.486 \times 10^{-11}$ | −3.112 |
| Rostralmiddlefrontal 12-lh | 0.344 | 0.8857 | $1.494 \times 10^{-11}$ | −3.120 |
| Rostralmiddlefrontal 12-rh | 0.336 | 0.7840 | $1.189 \times 10^{-11}$ | −3.210 |
| Rostralmiddlefrontal 13-rh | 0.34 | 0.8510 | $1.428 \times 10^{-11}$ | −3.220 |
| Superiorfrontal 1-lh | 0.336 | 0.9122 | $1.420 \times 10^{-11}$ | −3.101 |

**Appendix 1—table 2.** In the "Stay/Cue" choice, we require that the activity of more than 50% of brain regions remains significantly correlated with the value of reducing risk for over 0.152 seconds, with a significance p<0.05 (after false discovery rate correction).
The brain regions are delineated according to the "aparc sub" parcellation.

| Regressor | p-Value | Proportion | Duration | |
|---|---|---|---|---|
| Value of avoiding risk | 0.05 | 0.5 | 0.152 seconds | |
| **Brain region** | **Duration** | **Proportion** | **Regression coefficient** | ***t*-Value** |
| Caudalmiddlefrontal 5-lh | 0.152 | 0.9676 | $1.151 \times 10^{-11}$ | −3.363 |
| Caudalmiddlefrontal 6-lh | 0.156 | 0.9837 | $1.006 \times 10^{-11}$ | −3.307 |
| Insula 2-lh | 0.156 | 0.9597 | $1.270 \times 10^{-11}$ | −3.350 |
| Insula 3-lh | 0.156 | 0.9359 | $1.229 \times 10^{-11}$ | −3.251 |
| Medialorbitofrontal 5-lh | 0.164 | 0.8659 | $1.386 \times 10^{-11}$ | −3.081 |
| Parsopercularis 3-lh | 0.160 | 0.9479 | $1.450 \times 10^{-11}$ | −3.334 |
| Postcentral 10-lh | 0.160 | 0.9450 | $1.071 \times 10^{-11}$ | −3.054 |
| Postcentral 11-lh | 0.152 | 0.9737 | $1.111 \times 10^{-11}$ | −3.171 |

*Appendix 1—table 2 Continued on next page*

*Appendix 1—table 2 Continued*

| Regressor | p-Value | Proportion | Duration | |
|---|---|---|---|---|
| Postcentral 13-lh | 0.152 | 0.8991 | $1.190 \times 10^{-11}$ | −2.951 |
| Precentral 11-lh | 0.152 | 0.9799 | $1.033 \times 10^{-11}$ | −3.376 |
| Precentral 12-lh | 0.156 | 0.9630 | $1.170 \times 10^{-11}$ | −3.280 |
| Precentral 13-lh | 0.152 | 0.9868 | $1.033 \times 10^{-11}$ | −3.347 |
| Precentral 8-lh | 0.152 | 0.9757 | $1.028 \times 10^{-11}$ | −3.434 |
| Precentral 9-lh | 0.152 | 0.9649 | $9.692 \times 10^{-11}$ | −3.323 |
| Rostralanteriorcingulate 2-lh | 0.152 | 0.8816 | $1.007 \times 10^{-11}$ | −2.970 |
| Superiorfrontal 17-lh | 0.152 | 0.9868 | $1.001 \times 10^{-11}$ | −3.371 |
| Supramarginal 3-lh | 0.156 | 0.8521 | $1.260 \times 10^{-11}$ | −3.068 |

**Appendix 1—table 3.** In the "Stay/Cue" choice, we require that the activity of more than 50% of brain regions remains significantly correlated with the extrinsic value for over 0.128 seconds, with a significance p<0.05 (after false discovery rate correction).
The brain regions are delineated according to the "aparc sub" parcellation.

| Regressor | p-Value | Proportion | Duration | |
|---|---|---|---|---|
| Extrinsic value | 0.05 | 0.5 | 0.128 seconds | |
| Brain region | Duration | Proportion | regression coefficient | t-Value |
| Bankssts 1-rh | 0.136 | 0.9632 | $4.173 \times 10^{-11}$ | 3.547 |
| Bankssts 2-rh | 0.128 | 0.9583 | $3.646 \times 10^{-11}$ | 3.432 |
| Fusiform 7-rh | 0.136 | 0.9647 | $4.727 \times 10^{-11}$ | 3.806 |
| Inferiorparietal 9-rh | 0.132 | 0.9532 | $3.946 \times 10^{-11}$ | 3.560 |
| Inferiortemporal 4-rh | 0.132 | 0.9731 | $5.742 \times 10^{-11}$ | 3.786 |
| Inferiortemporal 5-rh | 0.14 | 0.9771 | $5.648 \times 10^{-11}$ | 3.917 |
| Inferiortemporal 6-rh | 0.136 | 1.0000 | $5.635 \times 10^{-11}$ | 3.651 |
| Inferiortemporal 7-rh | 0.136 | 0.9559 | $4.850 \times 10^{-11}$ | 3.510 |
| Middletemporal 1-rh | 0.132 | 0.9495 | $4.260 \times 10^{-11}$ | 3.326 |
| Middletemporal 3-rh | 0.14 | 0.9679 | $5.060 \times 10^{-11}$ | 3.521 |
| Middletemporal 4-rh | 0.128 | 0.9688 | $4.636 \times 10^{-11}$ | 3.564 |
| Middletemporal 5-rh | 0.132 | 0.9610 | $5.078 \times 10^{-11}$ | 3.490 |
| Middletemporal 6-rh | 0.164 | 0.9338 | $5.983 \times 10^{-11}$ | 3.673 |
| Middletemporal 7-rh | 0.128 | 0.8880 | $4.359 \times 10^{-11}$ | 3.242 |
| Superiortemporal 3-rh | 0.128 | 0.9886 | $3.652 \times 10^{-11}$ | 3.454 |
| Superiortemporal 4-rh | 0.14 | 0.9629 | $4.585 \times 10^{-11}$ | 3.494 |
| Superiortemporal 5-rh | 0.132 | 0.9865 | $4.22 \times 10^{-11}$ | 3.299 |
| Superiortemporal 6-rh | 0.132 | 0.9449 | $3.658 \times 10^{-11}$ | 3.325 |
| Superiortemporal 9-rh | 0.132 | 0.9596 | $3.435 \times 10^{-11}$ | 3.429 |

**Appendix 1—table 4.** In the result stage after the "Stay/Cue" choice, we require that the activity of more than 50% of brain regions remains significantly correlated with (the value of) avoiding risk for over 0.32 seconds, with a significance p<0.05 (after false discovery rate correction).
The brain regions are delineated according to the "aparc sub" parcellation.

| Regressor | p-value | Proportion | Duration | |
|---|---|---|---|---|
| (The value of) avoiding risk | 0.05 | 0.5 | 0.32 seconds | |
| Brain region | Duration | Proportion | regression coefficient | t-Value |
| Caudalanteriorcingulate 1-lh | 0.344 | 0.9341 | $1.523 \times 10^{-11}$ | −3.023 |
| Caudalanteriorcingulate 2-lh | 0.332 | 0.9375 | $1.458 \times 10^{-11}$ | −2.985 |
| Lateralorbitofrontal 1-rh | 0.328 | 0.8659 | $1.733 \times 10^{-11}$ | −2.869 |
| Medialorbitofrontal 5-lh | 0.376 | 0.9109 | $1.839 \times 10^{-11}$ | −3.001 |
| Middletemporal 1-lh | 0.324 | 0.8642 | $2.245 \times 10^{-11}$ | −2.944 |
| Middletemporal 5-lh | 0.344 | 0.9201 | $2.466 \times 10^{-11}$ | −3.098 |
| Parstriangularis 1-rh | 0.364 | 0.9194 | $1.804 \times 10^{-11}$ | −3.103 |
| Parstriangularis 2-rh | 0.344 | 0.8709 | $1.936 \times 10^{-11}$ | −3.038 |
| Parstriangularis 3-rh | 0.328 | 0.9079 | $1.893 \times 10^{-11}$ | −3.117 |
| Parstriangularis 4-rh | 0.364 | 0.9011 | $2.057 \times 10^{-11}$ | −3.278 |
| Rostralanteriorcingulate 2-lh | 0.368 | 0.9348 | $1.553 \times 10^{-11}$ | −3.095 |
| Rostralanteriorcingulate 2-rh | 0.328 | 0.9321 | $1.469 \times 10^{-11}$ | −2.847 |
| Rostralmiddlefrontal 1-rh | 0.344 | 0.9120 | $1.571 \times 10^{-11}$ | −3.043 |
| Rostralmiddlefrontal 10-rh | 0.328 | 0.9011 | $1.772 \times 10^{-11}$ | −3.080 |
| Rostralmiddlefrontal 12-rh | 0.36 | 0.9016 | $1.722 \times 10^{-11}$ | −3.102 |
| Rostralmiddlefrontal 4-rh | 0.348 | 0.8931 | $1.825 \times 10^{-11}$ | −2.994 |
| Rostralmiddlefrontal 5-rh | 0.328 | 0.9463 | $1.664 \times 10^{-11}$ | −3.000 |
| Rostralmiddlefrontal 8-rh | 0.336 | 0.9365 | $1.730 \times 10^{-11}$ | −3.070 |
| Superiorfrontal 5-rh | 0.332 | 0.9277 | $1.548 \times 10^{-11}$ | −2.948 |
| Superiorfrontal 6-rh | 0.328 | 0.9268 | $1.585 \times 10^{-11}$ | −2.966 |
| Superiortemporal 7-lh | 0.34 | 0.8882 | $2.055 \times 10^{-11}$ | −3.017 |

**Appendix 1—table 5.** In the "Safe/Risk" choice, we require that the activity of more than 90% of brain regions remains significantly correlated with the expected free energy for over 1.88 seconds, with a significance p<0.001 (after false discovery rate correction).
The brain regions are delineated according to the "aparc sub" parcellation.

| Regressor | p-Value | Proportion | Duration | |
|---|---|---|---|---|
| Expected free energy | 0.001 | 0.5 | 1.88 seconds | |
| Brain region | Duration | Proportion | Regression coefficient | t-Value |
| Caudalmiddlefrontal 2-lh | 1.908 | 0.9783 | $2.030 \times 10^{-11}$ | −4.746 |
| Insula 6-lh | 1.896 | 0.9579 | $2.223 \times 10^{-11}$ | −4.693 |
| Middletemporal 5-lh | 1.904 | 0.9850 | $3.350 \times 10^{-11}$ | −5.115 |
| Middletemporal 6-lh | 1.92 | 0.9753 | $3.676 \times 10^{-11}$ | −4.988 |
| Parsorbitalis 2-lh | 1.912 | 0.9187 | $2.694 \times 10^{-11}$ | −4.803 |
| Parstriangularis 1-lh | 1.884 | 0.9581 | $2.580 \times 10^{-11}$ | −4.717 |
| Parstriangularis 2-lh | 1.896 | 0.9768 | $2.619 \times 10^{-11}$ | −4.814 |
| Rostralmiddlefrontal 1-lh | 1.908 | 0.9830 | $2.243 \times 10^{-11}$ | −4.819 |
| Rostralmiddlefrontal 2-lh | 1.896 | 0.9783 | $2.216 \times 10^{-11}$ | −4.789 |

*Appendix 1—table 5 Continued on next page*

*Appendix 1—table 5 Continued*

| Regressor | p-Value | Proportion | Duration | |
|---|---|---|---|---|
| Rostralmiddlefrontal 4-lh | 1.884 | 0.9662 | $2.082 \times 10^{-11}$ | –4.716 |
| Rostralmiddlefrontal 6-lh | 1.904 | 0.9898 | $2.470 \times 10^{-11}$ | –4.901 |

**Appendix 1—table 6.** In the "Safe/Risk" choice, we require that the activity of more than 50% of brain regions remains significantly correlated with the value of reducing ambiguity for over 0.14 seconds, with a significance p<0.05 (after false discovery rate correction).
The brain regions are delineated according to the "aparc sub" parcellation.

| Regressor | p-Value | Proportion | Duration | |
|---|---|---|---|---|
| Value of reducing ambiguity | 0.05 | 0.5 | 0.14 seconds | |
| Brain region | Duration | Proportion | Regression coefficient | *t*-Value |
| Insula 6-lh | 0.144 | 0.9037 | $5.112 \times 10^{-11}$ | –3.165 |
| Lateralorbitofrontal 4-lh | 0.144 | 0.8782 | $4.612 \times 10^{-11}$ | –2.971 |
| Parsorbitalis 2-lh | 0.144 | 0.7381 | $5.591 \times 10^{-11}$ | –3.059 |
| Rostralmiddlefrontal 1-lh | 0.148 | 0.8730 | $4.907 \times 10^{-11}$ | –3.107 |
| Rostralmiddlefrontal 6-lh | 0.16 | 0.8679 | $5.244 \times 10^{-11}$ | –3.067 |
| Superiorfrontal 10-lh | 0.152 | 0.9046 | $4.974 \times 10^{-11}$ | –3.065 |
| Superiorfrontal 6-lh | 0.148 | 0.8845 | $4.779 \times 10^{-11}$ | –2.946 |

**Appendix 1—table 7.** In the "Safe/Risk" choice, we require that the activity of more than 50% of brain regions remains significantly correlated with the extrinsic value for over 1.68 seconds, with a significance p<0.001 (after false discovery rate correction).
The brain regions are delineated according to the "aparc sub" parcellation.

| Regressor | p-Value | Proportion | Duration | |
|---|---|---|---|---|
| Extrinsic value | 0.001 | 0.5 | 1.68 seconds | |
| Brain region | Duration | Proportion | Regression coefficient | *t*-Value |
| Caudalmiddlefrontal 2-lh | 1.712 | 0.9626 | $2.131 \times 10^{-11}$ | 4.607 |
| Caudalmiddlefrontal 3-lh | 1.692 | 0.9485 | $2.081 \times 10^{-11}$ | 4.523 |
| Lateralorbitofrontal 4-lh | 1.688 | 0.9437 | $2.271 \times 10^{-11}$ | 4.595 |
| Middletemporal 4-lh | 1.692 | 0.9063 | $3.402 \times 10^{-11}$ | 4.727 |
| Middletemporal 5-lh | 1.78 | 0.9559 | $3.490 \times 10^{-11}$ | 4.905 |
| Middletemporal 6-lh | 1.764 | 0.9558 | $3.889 \times 10^{-11}$ | 4.847 |
| Middletemporal 6-rh | 1.732 | 0.9301 | $2.865 \times 10^{-11}$ | 4.687 |
| Parsopercularis 2-lh | 1.74 | 0.9478 | $2.697 \times 10^{-11}$ | 4.695 |
| Parsopercularis 4-lh | 1.712 | 0.9528 | $2.647 \times 10^{-11}$ | 4.654 |
| Parsorbitalis 2-lh | 1.748 | 0.9059 | $2.849 \times 10^{-11}$ | 4.675 |
| Parstriangularis 1-lh | 1.732 | 0.9310 | $2.759 \times 10^{-11}$ | 4.655 |
| Parstriangularis 2-lh | 1.76 | 0.9591 | $2.798 \times 10^{-11}$ | 4.739 |
| Parstriangularis 3-lh | 1.704 | 0.9577 | $2.565 \times 10^{-11}$ | 4.673 |
| Precentral 13-lh | 1.696 | 0.9682 | $1.950 \times 10^{-11}$ | 4.744 |
| Precentral 14-lh | 1.74 | 0.9688 | $2.066 \times 10^{-11}$ | 4.799 |

*Appendix 1—table 7 Continued on next page*

*Appendix 1—table 7 Continued*

| Regressor | p-Value | Proportion | Duration | |
|---|---|---|---|---|
| Rostralmiddlefrontal 1-lh | 1.74 | 0.9664 | $2.351 \times 10^{-11}$ | 4.653 |
| Rostralmiddlefrontal 2-lh | 1.716 | 0.9693 | $2.357 \times 10^{-11}$ | 4.692 |
| Rostralmiddlefrontal 4-lh | 1.712 | 0.9541 | $2.217 \times 10^{-11}$ | 4.634 |
| Rostralmiddlefrontal 5-lh | 1.708 | 0.9653 | $2.333 \times 10^{-11}$ | 4.663 |
| Rostralmiddlefrontal 6-lh | 1.72 | 0.9867 | $2.598 \times 10^{-11}$ | 4.74 |
| Rostralmiddlefrontal 7-lh | 1.7 | 0.9277 | $2.221 \times 10^{-11}$ | 4.602 |
| Rostralmiddlefrontal 8-lh | 1.7 | 0.9788 | $2.520 \times 10^{-11}$ | 4.703 |

**Appendix 1—table 8.** In the result stage of the "Safe/Risk" choice, we require that the activity of more than 50% of brain regions remains significantly correlated with the extrinsic value for over 0.248 seconds, with a significance $p<0.05$ (after false discovery rate correction).
The brain regions are delineated according to the "aparc sub" parcellation.

| Regressor | p-Value | Proportion | Duration | |
|---|---|---|---|---|
| Extrinsic value | 0.05 | 0.5 | 0.248 seconds | |
| Brain region | Duration | Proportion | Regression coefficient | *t*-Value |
| Fusiform 3-rh | 0.252 | 0.9714 | $1.090 \times 10^{-11}$ | –3.202 |
| Fusiform 5-rh | 0.256 | 0.9375 | $9.409 \times 10^{-12}$ | –3.063 |
| Inferiorparietal 11-rh | 0.252 | 0.9538 | $8.761 \times 10^{-12}$ | –3.029 |
| Inferiorparietal 5-rh | 0.256 | 0.9473 | $9.393 \times 10^{-12}$ | –3.067 |
| Inferiortemporal 7-rh | 0.256 | 0.9703 | $1.157 \times 10^{-11}$ | –3.357 |
| Lateraloccipital 3-rh | 0.268 | 0.8806 | $9.523 \times 10^{-12}$ | –2.875 |
| Lateraloccipital 4-rh | 0.26 | 0.9538 | $8.228 \times 10^{-12}$ | –2.902 |
| Lateraloccipital 5-rh | 0.252 | 0.9224 | $9.960 \times 10^{-12}$ | –2.908 |
| Lateraloccipital 8-rh | 0.26 | 0.9790 | $9.966 \times 10^{-12}$ | –3.092 |
| Lateraloccipital 9-rh | 0.252 | 0.9302 | $9.413 \times 10^{-12}$ | –3.026 |
| Lingual 5-rh | 0.256 | 0.9792 | $1.097 \times 10^{-11}$ | –3.155 |
| Paracentral 2-rh | 0.252 | 0.9762 | $5.791 \times 10^{-12}$ | –3.065 |
| Parahippocampal 2-rh | 0.256 | 0.9896 | $1.031 \times 10^{-11}$ | –3.225 |
| Postcentral 10-rh | 0.252 | 0.9707 | $7.185 \times 10^{-12}$ | –2.997 |
| Precuneus 8-rh | 0.256 | 0.9398 | $6.078 \times 10^{-12}$ | –2.904 |
| Superiorparietal 11-rh | 0.26 | 0.9303 | $8.428 \times 10^{-12}$ | –2.895 |
| Superiorparietal 3-rh | 0.268 | 0.9091 | $7.188 \times 10^{-12}$ | –3.0 |
| Superiorparietal 6-rh | 0.264 | 0.9318 | $8.186 \times 10^{-12}$ | –3.01 |
| Superiorparietal 7-rh | 0.256 | 0.9420 | $7.757 \times 10^{-12}$ | –2.958 |
| Superiorparietal 8-rh | 0.264 | 0.9674 | $7.950 \times 10^{-12}$ | –2.989 |

**Appendix 1—table 9.** In the result stage of the 'Safe/Risk' choice, we require that the activity of more than 50% of brain regions remains significantly correlated with (the value of) reducing ambiguity for over 0.072 seconds, with a significance $p<0.05$.
The brain regions are delineated according to the "aparc sub" parcellation.

| Regressor | p-Value | Proportion | Duration | |
|---|---|---|---|---|
| (The value of) reducing ambiguity | 0.05 | 0.5 | 0.072 seconds | |
| **Brain region** | **Duration** | **Proportion** | **Regression coefficient** | *t-*Value |
| Paracentral 4-rh | 0.072 | 0.9697 | $2.809 \times 10^{-11}$ | –3.321 |
| Paracentral 5-rh | 0.072 | 0.9506 | $2.803 \times 10^{-11}$ | –3.156 |
| Paracentral 6-rh | 0.072 | 0.9778 | $2.420 \times 10^{-11}$ | –3.145 |
| Precentral 11-rh | 0.072 | 0.9861 | $3.115 \times 10^{-11}$ | –3.316 |
| Precentral 15-rh | 0.072 | 0.9596 | $2.952 \times 10^{-11}$ | –3.278 |
| Precentral 16-rh | 0.072 | 0.9293 | $3.079 \times 10^{-11}$ | –3.23 |
| Precentral 7-rh | 0.072 | 0.9242 | $2.994 \times 10^{-11}$ | –3.27 |
| Superiorparietal 3-rh | 0.072 | 0.9343 | $2.940 \times 10^{-11}$ | –3.132 |
| Superiorparietal 6-rh | 0.072 | 0.9611 | $3.410 \times 10^{-11}$ | –3.19 |
| Supramarginal 1-rh | 0.072 | 0.9615 | $3.942 \times 10^{-11}$ | –3.41 |
| Supramarginal 9-rh | 0.072 | 0.9213 | $3.731 \times 10^{-11}$ | –3.452 |

