## [Editor Report · eLife Assessment]

This **valuable** study addresses a central question in systems neuroscience (validation of active inference models of exploration) using a combination of behavior, neuroimaging, and modeling. The data provided offers **solid** evidence that humans do perceive, choose, and learn in a manner consistent with the essential ingredients of active inference, and that quantities that correlate with relevant parameters of this active inference scheme are encoded in different regions of the brain.

---

## [Referee Report · Reviewer #1 (Public review)]

Summary:

This paper presents a compelling and comprehensive study of decision-making under uncertainty. It addresses a fundamental distinction between belief-based (cognitive neuroscience) formulations of choice behavior with reward-based (behavioral psychology) accounts. Specifically, it asks whether active inference provides a better account of planning and decision making, relative to reinforcement learning. To do this, the authors use a simple but elegant paradigm that includes choices about whether to seek both information and rewards. They then assess the evidence for active inference and reinforcement learning models of choice behavior, respectively. After demonstrating that active inference provides a better explanation of behavioral responses, the neuronal correlates of epistemic and instrumental value (under an optimized active inference model) are characterized using EEG. Significant neuronal correlates of both kinds of value were found in sensor and source space. The source space correlates are then discussed sensibly, in relation to the existing literature on the functional anatomy of perceptual and instrumental decision-making under uncertainty.

Comments on revisions:

Many thanks for attending to my previous comments. I think your manuscript is now easier to read - and your new (Bayesian) analyses are described clearly.

---

## [Referee Report · Reviewer #3 (Public review)]

Summary:

This paper aims to investigate how the human brain represents different forms of value and uncertainty that participate in active inference within a free-energy framework, in a two-stage decision task involving contextual information sampling, and choices between safe and risky rewards, which promotes shifting between exploration and exploitation. They examine neural correlates by recording EEG and comparing activity in the first vs second half of trials and between trials in which subjects did and did not sample contextual information, and perform a regression with free-energy-related regressors against data "mapped to source space."

Strengths:

This two-stage paradigm is cleverly designed to incorporate several important processes of learning, exploration/exploitation and information sampling that pertain to active inference. Although scalp/brain regions showing sensitivity to the active-inference related quantities do not necessarily suggest what role they play, they are illuminating and useful as candidate regions for further investigation. The aims are ambitious, and the methodologies are impressive. The paper lays out an extensive introduction to the free energy principle and active inference to make the findings accessible to a broad readership.

Weaknesses:

It is worth noting that the high lower-cutoff of 1 Hz in the bandpass filter, included to reduce the impact of EEG noise, would remove from the EEG any sustained, iteratively updated representation that evolves with learning across trials, or choice-related processes that unfold slowly over the course of the 2-second task windows. It is thus possible there are additional processes related to the active inference quantities that are missed here. This is not a flaw as one must always try to balance noise removal against signal removal in filter settings - it is just a caveat. As the authors also note, the regions showing up as correlated with model parameters change depending on source modelling method and correction for multiple comparisons, warranting some caution around the localisation aspect.

---

## [Author Response]

The following is the authors’ response to the previous reviews.

**Reviewer #1 (Public Review):**
Summary:This paper presents a compelling and comprehensive study of decision-making under uncertainty. It addresses a fundamental distinction between belief-based (cognitive neuroscience) formulations of choice behavior with reward-based (behavioral psychology) accounts. Specifically, it asks whether active inference provides a better account of planning and decision making, relative to reinforcement learning. To do this, the authors use a simple but elegant paradigm that includes choices about whether to seek both information and rewards. They then assess the evidence for active inference and reinforcement learning models of choice behavior, respectively. After demonstrating that active inference provides a better explanation of behavioral responses, the neuronal correlates of epistemic and instrumental value (under an optimized active inference model) are characterized using EEG. Significant neuronal correlates of both kinds of value were found in sensor and source space. The source space correlates are then discussed sensibly, in relation to the existing literature on the functional anatomy of perceptual and instrumental decision-making under uncertainty.

We are deeply grateful for your careful review of our work and your suggestions. Your insights have helped us identify areas where we can strengthen the arguments and clarify the methodology. We hope to apply the idea of active inference to our future work, emphasizing the integrity of perception and action.

**Reviewer #1 (Recommendations For The Authors):**
Many thanks for attending to my previous suggestions. I think your presentation is now much clearer and nicely aligned with the active inference literature.There is one outstanding issue. I think you have overinterpreted the two components of epistemic value in Equation 8. The two components that you have called the value of reducing risk and the value of reducing ambiguity are not consistent with the normal interpretation. These two components are KL divergences that measure the expected information gain about parameters and states respectively.If you read the Schwartenbeck et al paper carefully, you will see that the first (expected information gain about parameters) is usually called novelty, while the second (expected information gain about states) is usually called salience.This means you can replace "the value of reducing ambiguity" with "novelty" and "the value of reducing risk" with "salience".For your interest, "risk" and "ambiguity" are alternative ways of decomposing expected free energy. In other words, you can decompose expected free energy into (negative) expected information gain and expected value (as you have done). Alternatively, you can rearrange the terms and express expected free energy as risk and ambiguity. Look at the top panel of Figure 4 in:
https://www.sciencedirect.com/science/article/pii/S0022249620300857
I hope that this helps.

We deeply thank you for your recommendations about the interpretation of the epistemic value in Equation 8. We have now corrected them to Novelty and Salience:

G(π,τ)=EQ~[ln⁡Q(A)−ln⁡P(A∣sτ,oτ,π) novelty lnln⁡Q(sτ∣π)−ln⁡Q(sτ∣oτ,π) salience ]⏟Epistemic value −EQ~[ln⁡P(oτ)]⏟Extrinsic value 

In addition, in order to avoid terminology conflicts with active inference and to describe these two different uncertainties, we replaced Ambiguity in the article with Novelty, referring to the uncertainty that can be reduced by sampling, and replaced Risk with Variability, referring to the uncertainty inherent in the environment (variance).

**Reviewer # 2 (Public Review):**
Summary**:**Zhang and colleagues use a combination of behavioral, neural, and computational analyses to test an active inference model of exploration in a novel reinforcement learning task..Strengths:The paper addresses an important question (validation of active inference models of exploration). The combination of behavior, neuroimaging, and modeling is potentially powerful for answering this question.I appreciate the addition of details about model fitting, comparison, and recovery, as well as the change in some of the methods.

We are deeply grateful for your careful review of our work and your suggestions. And we are also very sorry that in our last responses, there were a few suggestions from you that we did not respond them appropriately in our manuscript. We hope to be able to respond to these suggestions well in this revision. Thank you for your contribution to ensuring the scientificity and reproducibility of the work.

The authors do not cite what is probably the most relevant contextual bandit study, by Collins & Frank (2018, PNAS), which uses EEG.The authors cite Collins & Molinaro as a form of contextual bandit, but that's not the case (what they call "context" is just the choice set). They should look at the earlier work from Collins, starting with Collins & Frank (2012, EJN).

We deeply thank you for your comments. Now we add the relevant citations in the manuscript (line 46):

“These studies utilized different forms of multi-armed bandit tasks, e.g the restless multi-armed bandit tasks (Daw et al., 2006; Guha et al., 2010), risky/safe bandit tasks (Tomov et al., 2020; Fan et al., 2022; Payzan et al., 2013), contextual multi-armed bandit tasks (Collins & Frank, 2018; Schulz et al., 2015; Collins & Frank, 2012)”

Daw, N. D., O'doherty, J. P., Dayan, P., Seymour, B., & Dolan, R. J. (2006). Cortical substrates for exploratory decisions in humans. *Nature*, *441*(7095), 876-879.

Guha, S., Munagala, K., & Shi, P. (2010). Approximation algorithms for restless bandit problems. *Journal of the ACM (JACM)*, *58*(1), 1-50.

Tomov, M. S., Truong, V. Q., Hundia, R. A., & Gershman, S. J. (2020). Dissociable neural correlates of uncertainty underlie different exploration strategies. *Nature communications*, *11*(1), 2371.

Fan, H., Gershman, S. J., & Phelps, E. A. (2023). Trait somatic anxiety is associated with reduced directed exploration and underestimation of uncertainty. *Nature Human Behaviour*, *7*(1), 102-113.

Payzan-LeNestour, E., Dunne, S., Bossaerts, P., & O’Doherty, J. P. (2013). The neural representation of unexpected uncertainty during value-based decision making. *Neuron*, *79*(1), 191-201.

Collins, A. G., & Frank, M. J. (2018). Within-and across-trial dynamics of human EEG reveal cooperative interplay between reinforcement learning and working memory. *Proceedings of the National Academy of Sciences*, *115*(10), 2502-2507.

Schulz, E., Konstantinidis, E., & Speekenbrink, M. (2015, April). Exploration-exploitation in a contextual multi-armed bandit task. In *International conference on cognitive modeling* (pp. 118-123).

Collins, A. G., & Frank, M. J. (2012). How much of reinforcement learning is working memory, not reinforcement learning? A behavioral, computational, and neurogenetic analysis. *European Journal of Neuroscience*, *35*(7), 1024-1035.

Placing statistical information in a GitHub repository is not appropriate. This needs to be in the main text of the paper. I don't understand why the authors refer to space limitations; there are none for eLife, as far as I'm aware.

We deeply thank you for your comments. We calculated the average t-value of the brain regions with significant results over the significant time, and added the t-value results to the main text and supplementary materials.

In answer to my question about multiple comparisons, the authors have added the following: "Note that we did not attempt to correct for multiple comparisons; largely, because the correlations observed were sustained over considerable time periods, which would be almost impossible under the null hypothesis of no correlations." I'm sorry, but this does not make sense. Either the authors are doing multiple comparisons, in which case multiple comparison correction is relevant, or they are doing a single test on the extended timeseries, in which case they need to report that. There exist tools for this kind of analysis (e.g., Gershman et al., 2014, NeuroImage). I'm not suggesting that the authors should necessarily do this, only that their statistical approach should be coherent. As a reference point, the authors might look at the aforementioned Collins & Frank (2018) study.

We deeply thank you for your comments. We have now replaced all our results with the results after false discovery rate correction and added relevant descriptions (line 357,358):

“The significant results after false discovery rate (FDR) (Benjamini et al., 1995, Gershman et al., 2014) correction were shown in shaded regions. Additional regression results can be found in Supplementary Materials.”

Benjamini, Y., & Hochberg, Y. (1995). Controlling the false discovery rate: a practical and powerful approach to multiple testing. *Journal of the Royal statistical society: series B (Methodological)*, *57*(1), 289-300.

Gershman, S. J., Blei, D. M., Norman, K. A., & Sederberg, P. B. (2014). Decomposing spatiotemporal brain patterns into topographic latent sources. *NeuroImage*, *98*, 91-102.

After FDR correction, our results have changed slightly. We have updated our Results and Discussion section.

It should be acknowledged that the changes in these results may represent a certain degree of error in our data (perhaps because the EEG data is too noisy or because of the average template we used, ‘fsaverage’). Therefore, we added relevant discussion in the Discussion section (line527-529):

“It should be acknowledged that our EEG-based regression results are somewhat unstable, and the brain regions with significant regression are inconsistent before and after FDR correction. In future work, we should collect more precise neural data to reduce this instability.”

I asked the authors to show more descriptive comparison between the model and the data. Their response was that this is not possible, which I find odd given that they are able to use the model to define a probability distribution on choices. All I'm asking about here is to show predictive checks which build confidence in the model fit. The additional simulations do not address this. The authors refer to figures 3 and 4, but these do not show any direct comparison between human data and the model beyond model comparison metrics.

We deeply thank you for your comments. We now compare the participants’ behavioral data and the model’s predictions trial by trial (Figure 5). We can clearly see the participants’ behavioral strategies in different states and trials and the model’s prediction accuracy. We have added the discussion related to Figure 5 (line 309-318):

“Figure 5 shows the comparison between the active inference model and the behavioral data, where we can see that the model can fit the participants behavioral strategies well. In the “Stay-Cue" choice, participants always tend to choose to ask the ranger and rarely choose not to ask. When the context was unknown, participants chose the “Safe" option or the “Risky" option very randomly, and they did not show any aversion to variability. When given “Context 1", where the “Risky" option gave participants a high average reward, participants almost exclusively chose the “Risky" option, which provided more information in the early trials and was found to provide more rewards in the later rounds. When given “Context 2", where the “Risky" option gave participants a low average reward, participants initially chose the “Risky" option and then tended to choose the “Safe" option. We can see that participants still occasionally chose the “Risky" option in the later trials of the experiment, which the model does not capture. This may be due to the influence of forgetting. Participants chose the “Risky" option again to establish an estimate of the reward distribution.”

**Reviewer # 2 (Recommendations For The Authors):**
In the supplement, there are missing references ("[?]").

Thank you very much for pointing out this. We have now fixed this error.

**Reviewer # 3 (Public review):**
Summary:This paper aims to investigate how the human brain represents different forms of value and uncertainty that participate in active inference within a free-energy framework, in a two-stage decision task involving contextual information sampling, and choices between safe and risky rewards, which promotes shifting between exploration and exploitation. They examine neural correlates by recording EEG and comparing activity in the first vs second half of trials and between trials in which subjects did and did not sample contextual information, and perform a regression with free-energy-related regressors against data "mapped to source space."Strengths:This two-stage paradigm is cleverly designed to incorporate several important processes of learning, exploration/exploitation and information sampling that pertain to active inference. Although scalp/brain regions showing sensitivity to the active-inference related quantities do not necessary suggest what role they play, they are illuminating and useful as candidate regions for further investigation. The aims are ambitious, and the methodologies impressive. The paper lays out an extensive introduction to the free energy principle and active inference to make the findings accessible to a broad readership.Weaknesses:In its revised form the paper is complete in providing the important details. Though not a serious weakness, it is important to note that the high lower-cutoff of 1 Hz in the bandpass filter, included to reduce the impact of EEG noise, would remove from the EEG any sustained, iteratively updated representation that evolves with learning across trials, or choice-related processes that unfold slowly over the course of the 2-second task windows.

We are deeply grateful for your careful review of our work and your suggestions. We are very sorry that we did not modify our filter frequency (it would be a lot of work to modify it). Thank you very much for pointing this out. We noticed the shortcoming of the high lower-cutoff of 1 Hz in the bandpass filter. We will carefully consider the filter frequency when preprocessing data in future work. Thank you very much!